# Recent co-evolution of two pandemic plant diseases in a multi-hybrid swarm

Mostafa Rahnama [1,7], Bradford Condon[1], João P. Ascari[2], Julian R. Dupuis [3], Emerson M. Del Ponte [2], Kerry F. Pedley[4], Sebastián Martinez [5], Barbara Valent [6] & Mark L. Farman [1] ✉

Most plant pathogens exhibit host specificity but when former barriers to infection break down, new diseases can rapidly emerge. For a number of fungal diseases, there is increasing evidence that hybridization plays a major role in driving host jumps. However, the relative contributions of existing variation versus new mutations in adapting to new host(s) is unclear. Here we reconstruct the evolutionary history of two recently emerged populations of the fungus *Pyricularia oryzae* that are responsible for two new plant diseases: wheat blast and grey leaf spot of ryegrasses. We provide evidence that wheat blast/grey leaf spot evolved through two distinct mating episodes: the first occurred ~60 years ago, when a fungal individual adapted to *Eleusine* mated with another individual from *Urochloa*. Then, about 10 years later, a single progeny from this cross underwent a series of matings with a small number of individuals from three additional host-specialized populations. These matings introduced non-functional alleles of two key host-specificity factors, whose recombination in a multi-hybrid swarm probably facilitated the host jump. We show that very few mutations have arisen since the founding event and a majority are private to individual isolates. Thus, adaptation to the wheat or *Lolium* hosts appears to have been instantaneous, and driven entirely by selection on repartitioned standing variation, with no obvious role for newly formed mutations.

Many new plant diseases emerge when pathogen populations specialized on one or more other host plants overcome genetic barriers that once prevented infection of a former non-host. It is generally believed that new diseases start with a host jump, followed by an adaptive period that results in a gradual increase of pathogen fitness and dispersal[1–4], and this adaptive process is thought to be a major driver of pathogen diversification[4]—a process known as 'host jump speciation'[5]. For phytopathogenic fungi, however, interspecific hybridization and admixture are emerging as common themes in the evolution and spread of new diseases[6] and these processes alone can produce genetic radiations. Consequently, the relative contributions of pre-jump, standing variation versus post-jump, adaptive mutations in driving pathogen diversification are often unclear. Factors that can contribute to this uncertainty are knowledge gaps about a hybrid's immediate ancestors[7,8] or if recombination events were ancient and/or repeated over many years[9]. At the same time, the genetic basis for host specificity

[1]Department of Plant Pathology, University of Kentucky, Lexington, KY, USA. [2]Departamento de Fitopatologia, Universidade Federal de Viçosa, Viçosa-MG, Brazil. [3]Department of Entomology S-225 Agricultural Science Center, University of Kentucky, Lexington, KY, USA. [4]USDA/ARS/Foreign Disease Weed Science Research Unit, Fort Detrick, Frederick, MD, USA. [5]Laboratorio de Patología Vegetal, Instituto Nacional de Investigación Agropecuaria, Treinta y Tres, Uruguay. [6]Department of Plant Pathology, Kansas State University, Manhattan, KS, USA. [7]Present address: Department of Biology, Tennessee Tech University, Cookeville, TN, USA. ✉e-mail: farman@uky.edu

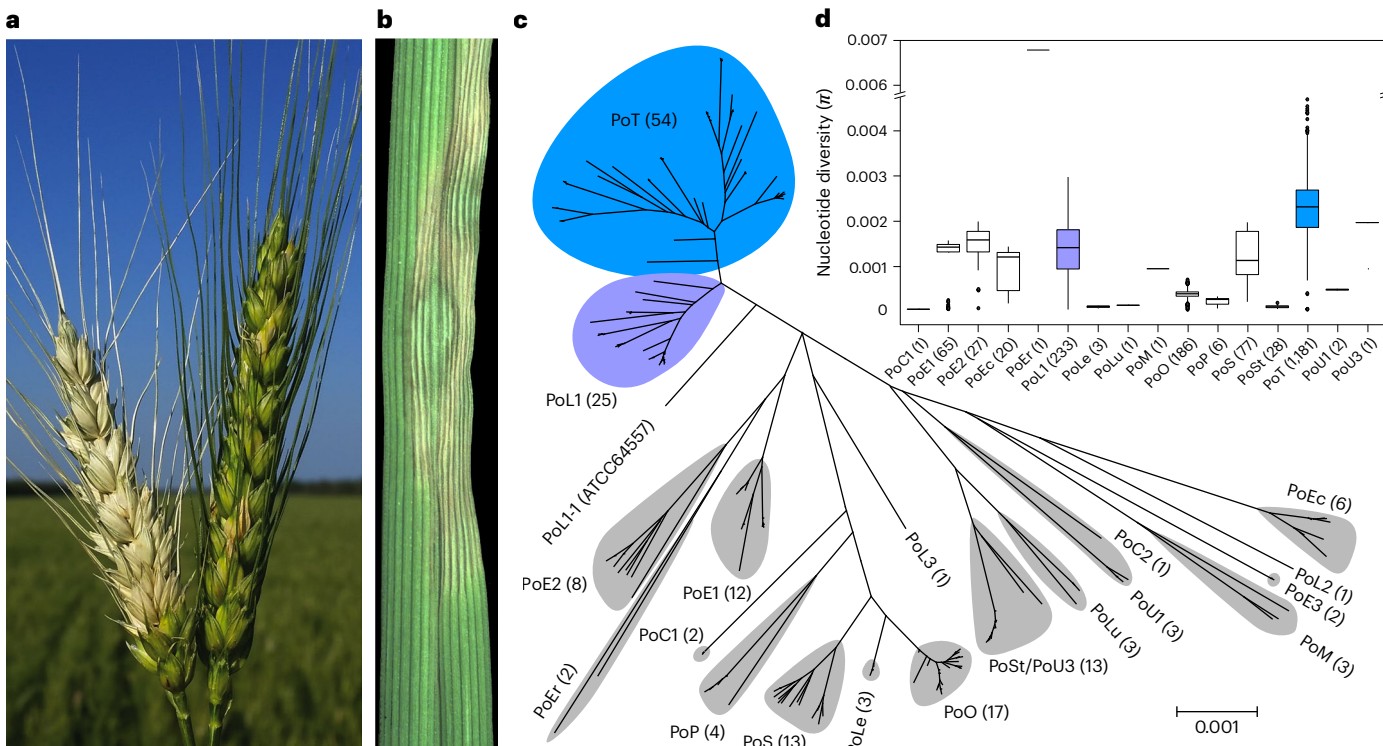

**Fig. 1 | Phylogenetic relationships between *P. oryzae* host-specialized populations. a**, Blasted wheat head (left) and healthy head (right). **b**, GLS on tall fescue (*Lolium arundinaceum*). **c**, Phylogenetic tree showing pairwise distances between isolates, and it is essentially the distance tree in reference[24], minus tip labels. Phylogenetic clades (lineages) are labelled according to genus of host, with numerical suffixes identifying distinct lineages that infect the same host. Naming scheme (lineage prefix, host) is as follows: PoC, *Cynodon*; PoE, *Eleusine*; PoEc, *Echinocloa*; PoEr, *Eragrostis*; PoL, *Lolium*; PoLe, *Leersia*; PoLu,

*Luziola*; PoM, *Melinis*; PoP, *Panicum*; PoS, *Setaria*; PoSt, *Stenotaphrum*; and PoU, *Urochloa*. Numbers of strains in each lineage are indicated in parentheses. Individual lineage members are listed in Supplementary Data File 1a. Grey shaded areas show DAPC groupings of the phylogenetic lineages (see below). **d**, Boxplot showing within-population pairwise nucleotide diversity (*π*). Population designations are shown on *x* axis and values in parentheses shown are the number of genetically independent samples (*n*). Bars, median; box, interquartile range (IQR); whiskers, smallest/largest value within the 1.5 times respective IQR limit.

is poorly understood for many fungal pathogens, so the mechanisms by which hybridization promotes host jumps and range expansions are unknown.

The wheat blast (WB) and grey leaf spot (GLS) diseases caused by the haploid fungus *Pyricularia oryzae* (synonymous with *Magnaporthe oryzae*) each caused major epidemics shortly after their detection. The initial outbreak on wheat (*Triticum aestivum*) (Fig. 1a) occurred in 1985 in Paraná, Brazil[10]. By 1990, it was present in all wheat-growing regions and soon thereafter in neighbouring countries[11–13]. Recent outbreaks in Asia and Africa[14] make WB an emerging concern for global agriculture. GLS was first detected in 1991 on perennial ryegrass (*Lolium perenne*) (Fig. 1b) in Pennsylvania[15]. By 1997, it had caused major epidemics throughout the central and eastern United States[16] and was present in Japan[17]. Prior incidences of *P. oryzae* on annual ryegrass (*Lolium multiflorum*) in Mississippi and Louisiana occurred in late 1971 (refs. 18,19) but initially appeared unrelated to the later outbreaks because the disease was localized and sporadic.

*P. oryzae* is best known as a global pathogen of rice (*Oryza sativa*) but also infects other crops such as oats, barley, millets, as well as a range of turf and weedy grasses[20]. Normally, the fungus is highly host specific and, although infection of multiple hosts is often seen in inoculation assays[21], the phylogenetic clustering of isolates by host of origin[22–24] (Fig. 1c) indicates that natural cross-infection is rare. Curiously, the newly emerged *P. oryzae* Triticum- and *Lolium*-adapted lineages (PoT and PoL1, respectively) appear to be less host restricted, with members having been found on 11 other Gramineae[23,25–27] (Extended Data Fig. 1a), which has prompted the suggestion that WB lacks specificity[27].

PoT and PoL1 exhibit far greater nucleotide diversity than other host-specialized populations of *P. oryzae*, as evidenced by phylogenetic branch lengths (Fig. 1c) that greatly exceed those seen among pathogens of rice (PoO) or foxtails (*Setaria spp.*, PoS)[24], whose estimated existences span hundreds and thousands of years, respectively[28,29]. High genetic and phenotypic diversity is often associated with recombining populations. However, in the pathogenic phase, *P. oryzae* propagates primarily through the production of asexual spores and the sexual phase has never been observed in nature. For this reason, field populations are usually considered asexual[30], although recent studies implicate the sexual cycle in the structuring of some host-specialized populations, including WB and GLS[23,31].

The prevailing model for WB evolution holds that it involved sequential loss/mutation of key host specificity genes, which allowed an *Eleusine* pathogen to jump hosts—first onto *Lolium*, then onto wheat lacking the *RWT3* resistance gene and, finally, onto *RWT3* wheat[32,33]. However, this could not explain how such high levels of genetic diversity arose in a population that had no obvious existence before 1985. In this Article, accordingly, we reasoned that understanding the high genetic diversity in WB/GLS would provide important new insights into their evolution.

## Results

### The WB genome is extensively admixed

In seeking to explain the high nucleotide diversity, we recalled that PoT/PoL1 possess up to four alleles for several phylogenetic markers, with each allele being identical to one found in another host-specialized population[23]. This pointed to the introgression of DNA from several

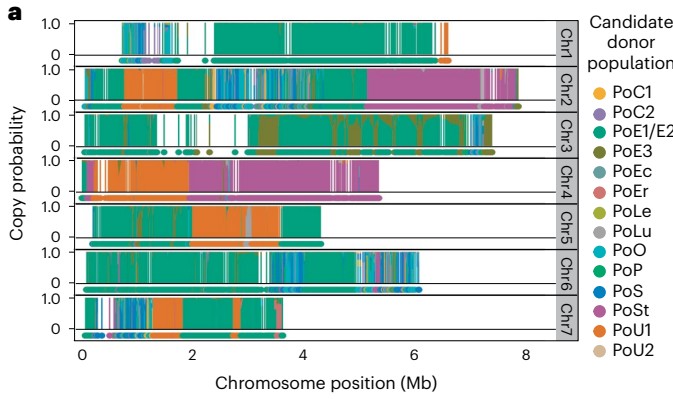

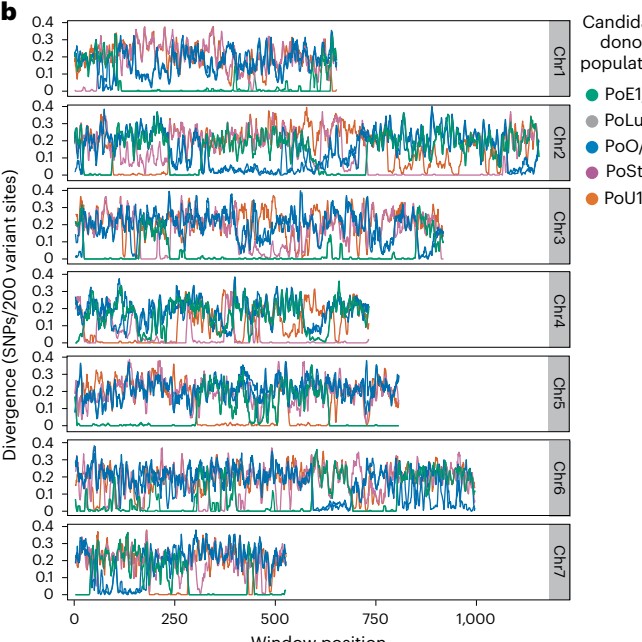

**Fig. 2 | The genome of the WB reference strain consisting of mosaic of sequences inherited from at least five pre-existing host-specialized lineages.** **a**, Chromopaintings of B71 chromosomes (chr) with stacked plots show copy probabilities for all candidate donors as listed in the legend. Regions with no data points correspond to repeated sequences or those exhibiting presence/absence of polymorphism. The condensed plots below each stack show the most probable donor for SNPs with high-confidence ancestry calls (probability, 2× runner-up). **b**, ShinyHaplotypes plots show divergence (SNPs/variant site) between B71 and a representative strain from each suspected donor lineage. Note that the *x* axis shows window number, because repeats and structural variation result in gaps that cause plots to render poorly when chromosome position is used.

sources. We used 20 kb of sequence surrounding two markers showing the highest diversity to perform phylogenetic analyses and, in both cases, four alleles were identified and each showed perfect/near-perfect nucleotide identity to an allele from another *P. oryzae* lineage. CH7BAC7 matched alleles found in lineages infecting *Urochloa* (PoU1 and PoU3), while *MPG1* identified with alleles found in *Eleusine* pathogens (PoE1). Both loci also showed alleles most closely related to those in PoO (*Oryza*), PoS (*Setaria*) and PoP (*Panicum*) (Extended Data Fig. 2). This suggested that PoT/PoL1 contains sequences donated by at least four host-specialized populations, with multiple incongruencies between the two trees pointing to abundant recombination between introgressed alleles.

We then surmised that the PoT/PoL1 genomes might be assembled entirely from sequences inherited from other host-specialized forms of

the fungus. To test this, we sought to ascertain the origins of genomic sequences in the WB reference isolate, B71[34]. First, genome-wide single nucleotide polymorphism (SNP) data were acquired for 96 candidate donor isolates (PoT/PoL1 excluded) that were assigned to 16 discrete populations based on discriminant analysis of principle components[35] (Extended Data Fig. 3 and Supplementary Data File 1a). When variant sites in the B71 genome were painted according to the probabilities they were donated by each of the 16 populations, the chromosomes were revealed to comprise blocks of contiguous sequences, with each having a high probability (>95%) of having been acquired from one of five distinct 'donor' populations (Fig. 2a). Importantly, the primary inferred donors precisely matched those previously predicted to be the sources of the introgressed CH7BAC7 and *MPG1* alleles, and there were no major segments whose copy probabilities were equally distributed among populations, which would be the case if any portions of the B71 genome had 'PoT-unique' heritage.

Next, we used a sliding window approach to scan the B71 chromosomes and measure haplotype divergence (number of SNPs/variant site) between B71 and a representative isolate from each of the candidate donor populations. The resulting plots revealed that the isolate with the lowest divergence relative to B71 often showed perfect haplotype identity (sequence identity) over multiple consecutive windows (average window size of 29.2 ± 2.4 kb, 95% confidence interval (CI)) (Fig. 2b). Overall, the proportion of windows showing perfect identity to at least one comparator ranged from 78% (on Chr7) to 96% (Chr1). Moreover, in many cases, the 'closest' isolate(s) was more similar to B71 than it was to other members of its own population (Extended Data Fig. 4). The only reasonable explanation for this pattern is that B71 recently inherited chromosome segments from members of the different host-specialized populations, which had former histories of divergence themselves. Here, the abrupt and reciprocal shifts in haplotype divergence that occur as comparisons progress along the chromosomes signal the traversal of analysis windows over crossovers between segments with different heritages.

The B71 chromosome segments showing the highest similarity to the PoO and PoS populations routinely showed higher divergence relevant to the best donor candidate, than did the rest of the B71 genome (Fig. 2b). Considering the foregoing data, we conclude that the respective chromosome segments were contributed by an unsampled population (hereafter known as 'PoX') that is closely related to, yet slightly diverged from, PoO/PoS.

## Haplotype analysis of PoT/PoL1 population members

To understand the unusually high nucleotide diversity within PoT/PoL1, we use sliding window haplotype comparisons to compare B71 with 116 other PoT/PoL1 isolates. As expected, this revealed many chromosome regions where B71 showed substantial divergence relative to other PoT/PoL1 members. However, these regions were unevenly distributed across the genome because long stretches of chromosomes 1, 2, 4, 5 and 6, spanning a total of 7 Mb (~15% of the genome), showed almost perfect identity across all isolates (Fig. 3). Importantly, even in the regions of high diversity, B71 still showed near-perfect identity to other strains, with plot lines hugging the *x* axis. This told us that all chromosome segments—no matter their origin—have accumulated similar (and very low) numbers of mutations since the population's foundation. This was true also for the regions with PoO/PoS-like haplotypes and, therefore, confirms the existence of an unsampled PoX donor.

The logical extension from the above findings is that nearly all of the nucleotide diversity within PoT/PoL1 must have arisen through the variable repartitioning of standing variation. More specifically, this points to a scenario in which PoT/PoL1 arose in a hybrid swarm, where a series of admixture events introduced genetic material from several formerly diverged donor populations. Matings between swarm members then led to extensive recombination and re-assortment of the various contributions, while still retaining substantial portions

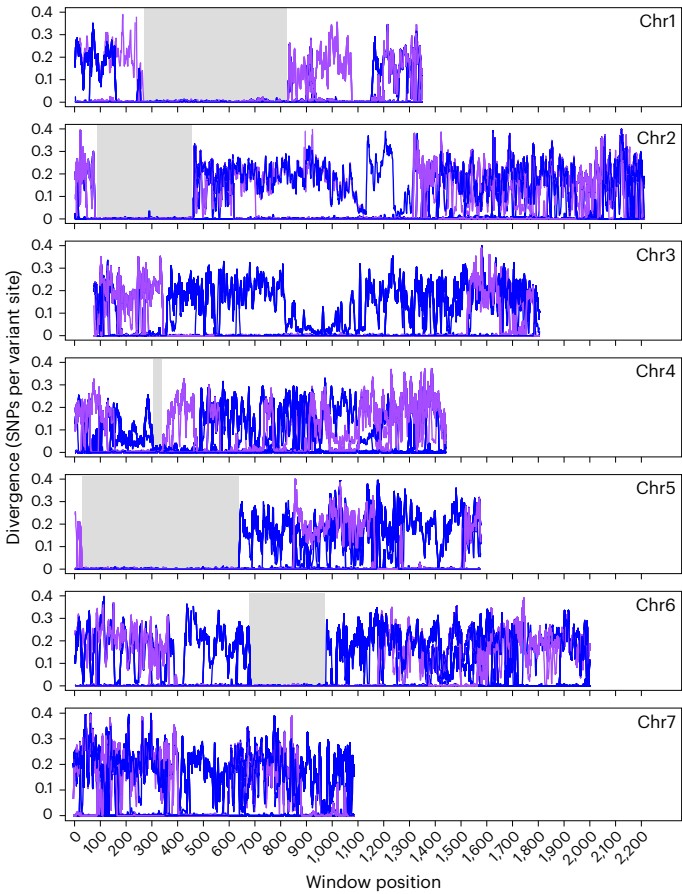

**Fig. 3 | Haplotype comparisons between B71 and 112 isolates from the PoT and PoL1 lineages.** Divergences relative to PoT are plotted in blue (plot lines, *n* = 86) and PoL in purple (*n* = 36). Note that extensive haplotype similarity among sets of isolates gives the appearance of single plots. This is most evident in regions on chromosomes (chr) 1, 2, 4, 5, and 6, where all isolates show near-perfect haplotype identity (grey shading).

of each donor genome within the newfound population. To test this 'multi-hybrid' swarm hypothesis, we first used SplitsTree to explore recombination within the PoT and PoL1 lineages. The resulting network contained deep reticulations that pointed to extensive recombination among population members (Extended Data Fig. 1b). Isolates grouped into clades comprising clones with very low genome-wide divergence, or they formed singleton branches. Reasoning that clonal isolates are unlikely to show major differences in patterns of admixture inheritance, we selected a single isolate from each of 43 clades (12 for PoL1 and 31 for PoT, Supplementary Data File 1b) for chromosome painting. This revealed that each clade/singleton had a distinct chromosomal haplotype that was defined by the specific combination of donor segments it inherited (Fig. 4a). It was also revealed that PoL1 members contained sizable contributions of DNA from a fifth lineage found on *Luziola* sp. (PoLu), with most of these sequences being present on chromosome 4.

Hybrid swarms are characterized as populations in which swarm members undergo extensive intermating, as well as backcrossing to their parents[36]. Inspection of the chromopaintings revealed clear evidence of recombination/re-assortment among hybrid individuals from both the PoT and PoL1 populations, with the first indication being the widespread sharing of specific crossovers among population members. In the most extreme cases, divergent PoT/PoL1-1 haplotypes shared entire recombinant chromosomes (for example, chromosome 1 in PoL1-3 and PoT4, and chromosome 5 in PoL1-4 and PoT7) (Fig. 4a).

Second, reciprocal products of individual crossover events were identified in different haplotypic backgrounds (Extended Data Table 1)—a strong indication of genetic contributions by sibling progeny from single meiotic events. Third, there was evidence that several chromosomal haplotypes arose via backcrossing (see below). A consequence of this extensive recombination/re-assortment of different donor contributions was extensive variation in the relative proportions of chromosome ancestry among haplotypes (Extended Data Fig. 5 and Extended Data Table 2). Particularly striking examples are PoT members that inherited entire chromosomes from different donors (for example, chromosome 4 in PoT1 versus PoT26, and chromosome 7 in PoT1 versus PoT9). Surprisingly, there were also very few chromosome segments with shared heritage across all members of PoL1 or PoT, or among strains found on *Lolium* or wheat (Fig. 4b).

### Analysis of 'early' strains identified two key founders

*P. oryzae* mates via hyphal fusion between strains of opposite mating type. Therefore, to explain the chromosomal constitutions of PoT/PoL1, we hypothesized that evolution must have occurred in a stepwise fashion, starting with a mating between a pair of fungal individuals from two of the formerly diverged donor lineages. A recombinant individual from this founding haplotype would have then mated with a member of a third diverged lineage, and so on. Furthermore, because chromosome segments with common ancestry showed so little variation within the new population, we surmised that the founding cross must have occurred very recently, so that isolates collected soon after the first disease outbreak might provide insights into early population structure and could be intermediate admixtures. Most PoL1 isolates were collected during the mid-1990s epidemics, except for ATCC64557 (PoL1 haplotype 1; PoL1-1), which was isolated in 1980. Chromopainting revealed ATCC64557 to be a two-way admixture (Fig. 4a) and, based on the proportional inheritance of donor DNAs (~65:35, PoE1:PoU1) (Extended Data Fig. 5 and Extended Data Table 2), it is probably an F1 or BC1 progeny of a mating between strains from the PoE1 and PoU1 populations. Critically, with the exception of a few secondary PoE1 introgressions in some haplotypes (see below), the PoE1 and PoU1 segments present in ATCC64557 account for all PoE1 and PoU1 DNA in the PoT/PoL1 population. This, along with the absolute conservation of certain crossover points and the extremely small number of SNPs in chromosome segments inherited from the PoE1/PoU1 donors (Fig. 4a, and see below), leads us to conclude that the entire PoT/PoL1 population originated with a single progeny from the founding cross (hereafter termed admix 1).

Subsequent steps in the swarm's formation were illuminated by our access to a number of previously uncharacterized WB isolates collected between 1985 and 1989. Two of the 'earliest' strains from wheat, T47-3 (from the first outbreak in 1985) and T3-1 (1986), appear to be clones of one another and represent the PoT1 haplotype. PoT1 is a three-way admixture, with about 27% of the genome having come from PoX, and the remainder essentially coming from PoL1-1 (Fig. 4a). The PoX introgressions in PoT1 account for all PoX DNA found in PoT members, and ~65% of the PoX sequences in PoL1. Moreover, all of the crossovers between the PoE1, PoU1 and PoX segments in PoT1 are widely distributed among the other PoT/PoL1 haplotypes. Thus, PoT1 is a second key founder individual for WB/GLS evolution and appears to have arisen as an F1 or BC1 progeny of a mating between a member of the PoL1-1 founder haplotype and a PoX individual (admix 2).

Several PoL1 members contained PoE1 introgressions that are also not present in PoT1 and were presumably acquired by backcrossing with PoL1-1. A good example is a diagnostic PoE1 to PoU1 crossover that was regained on the left arm of chromosome 7 (Fig. 4a). Other such segments on chromosomes 5 and 7 could not have come from backcrosses because PoL1-1 had PoU1 heritage in those regions. Phylogenetic and haplotype analysis of representative sequences identified two distinct

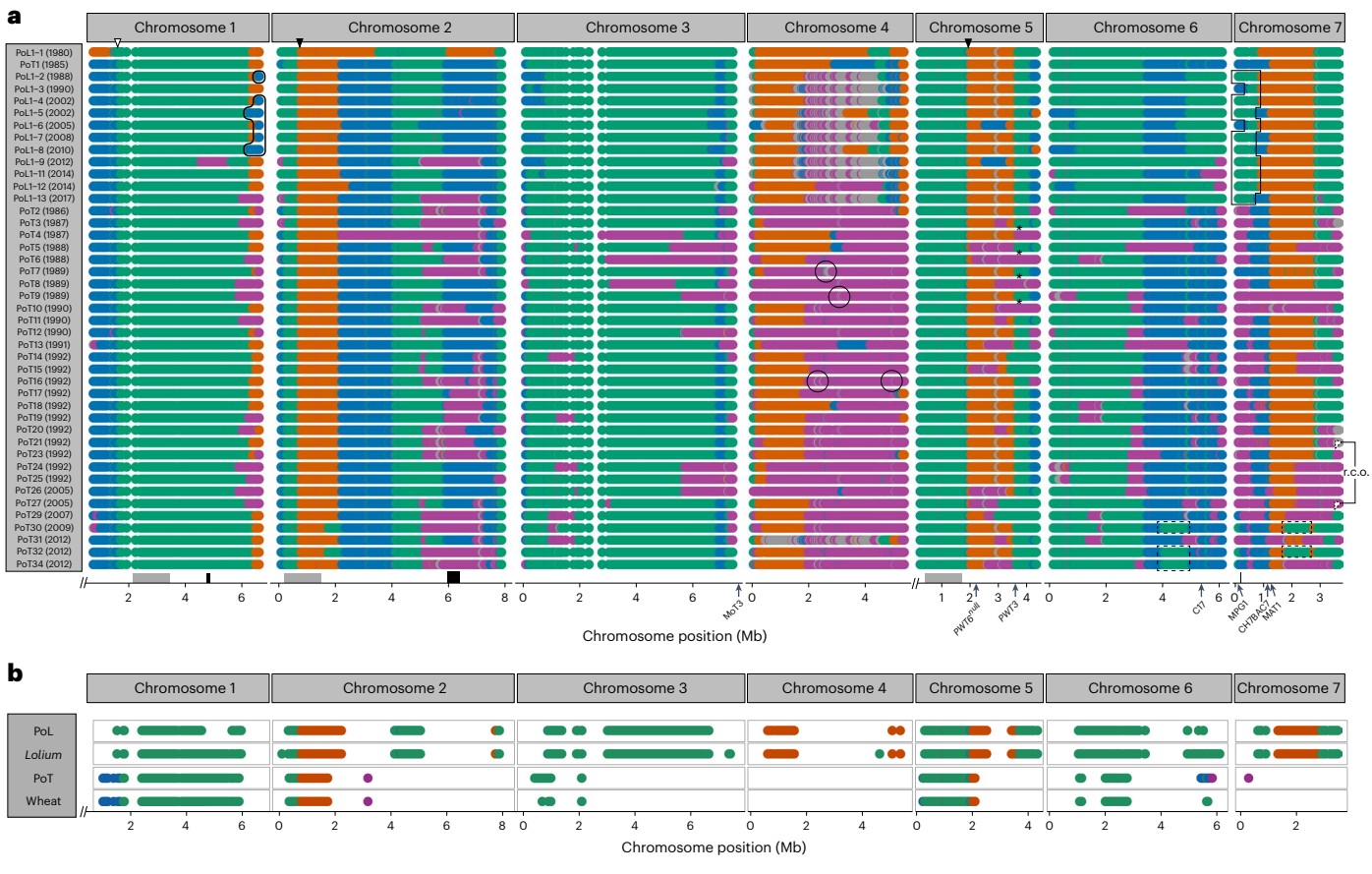

**Fig. 4 | Chromopaintings showing most probable donor. a**, Representative members of each identified PoT/PoL1 haplotype for which a high-quality assembly was available are shown. Only contributions from the five main donor lineages are shown, because predicted donations from other lineages were minor (<1%), rare and/or low confidence. Highlighted are examples of chromosome segments predicted to have been acquired: (i) from a PoLu donor, (ii) via backcrossing to PoL1 (black outlines with square corners), (iii) from a second PoX donor (rounded corners), and (iv) from a second PoE1 donor (outline with dotted lines). Black and white arrowheads show crossovers inherited from PoL1-1 and PoT1, respectively, and arrowheads with dotted outlines show reciprocal crossovers. Arrows along the *x* axis mark the locations of key loci. Also shown are regions on chromosomes 1, 2 and 5 used in tip dating (grey boxes) and analysis of secondary donations (black). **b**, Chromosome segments that exhibit shared ancestry among all members of the respective groups (PoL1; isolates from *Lolium*; PoT; and isolates from wheat) are shown. Predicted donors are shown in the legend at the bottom. PoE = PoE1 + E2 + E3 and PoX = PoEc + PoO + PoP + PoS.

PoE1 haplotypes (Extended Data Fig. 6) and, therefore, confirmed the occurrence of a secondary contribution (2PoE1). Likewise, the variable presence of short PoLu introgressions on a PoSt background in chromosome 4 of some PoT isolates (circled) signalled contributions from a second PoSt donor (2PoSt). Analysis of chromosome 7 sequences that appeared to have a shared PoSt ancestry identified two haplotypes (Extended Data Fig. 6), which confirmed a second contribution and also explained why the PoT population contains ~75% of the PoSt genome (Extended Data Fig. 5). Finally, some PoL1 haplotypes contained PoX sequences that were not present in PoT1 and must, therefore, have come from a sibling progeny of admix 2, or other member(s) of the PoX donor population. Analysis of a secondary PoX introgression on PoL1-6 chromosome 2 that overlapped with another PoX segment originally acquired in admix 2 confirmed the former scenario, because two distinct PoX haplotypes were found within the overlap and these resolved to a single haplotype after a presumed crossover point (Extended Data Fig. 6b). The high level of resolution provided by haplotype analysis also identified a number of extended gene conversion tracts in some isolates (Extended Data Fig. 6b). Apparent secondary PoU1 contributions on chromosome 5 in PoL1-4 and PoL1-8 are questionable because they are supported by just one data point in a region of uncertain heritage.

## Molecular dating of key events in WB/GLS evolution

To ensure that dating estimates were not influenced by the standing variation that predated WB/GLS's evolution, we used a carefully curated SNP dataset derived from chromosome regions that all analysed isolates appeared to have inherited from the PoL1-1 founder. We included data from three chromosomes (1, 2, and 5; 3.8 Mb total) to increase our chances of detecting recombination within the dataset. A total of 422 high-confidence SNPs were identified among 73 isolates, and these had an acceptable temporal signal (Fig. 5a) that was robust to bootstrapping (Fig. 5b). A generalized stepping-stone analysis provided marginally better support for a relaxed clock model (log Bayes factor 1), which yielded estimates of $1.27 \times 10^{-7}$ SNPs per site per year for the average nucleotide substitution rate and a time to most recent common ancestor (TMRCA) of ~50 years (Fig. 5c)—effectively dating admix 1 to about 1968 (95% CI 1952–1975) (Fig. 6). Values obtained under the strict clock model were very similar ($1.19 \times 10^{-7}$ SNPs per site per year; dated to 1966, 95% CI 1954–1974). These estimates should not have been greatly affected by recombination because only 15 of the 422 SNPs (3.5%) were shared between different haplotypes (Supplementary Data File 2).

Of the 31 PoL1/PoT haplotypes represented in the tree, 18 are rooted at nodes dated between 1978 and 1980 (Fig. 5c)—a pattern that is consistent with their emergence in a short-lived swarm, whose peak

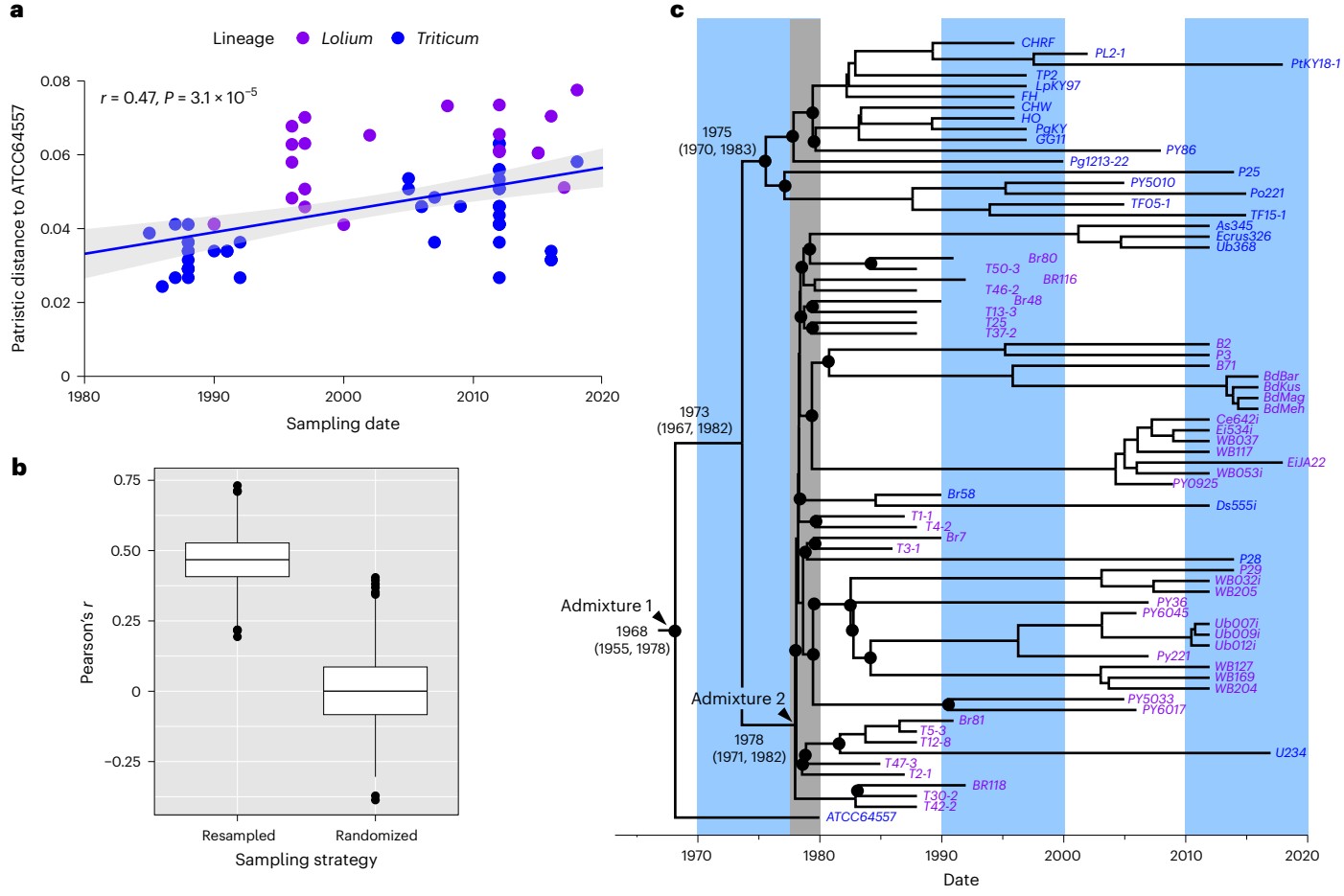

**Fig. 5 | Molecular dating of key events in PoT/PoL1 evolution. a**, Phylogenetic signal based on correlation between patristic distance versus sampling date was measured using Pearson's $r$ statistic ($n = 72$). Grey area shows the 95% CI of mean patristic distance at a given sampling time. **b**, Robustness of signal was determined by bootstrapping versus randomization of sampling dates (number of replicates is 1,000). Bars, median; box, IQR. Test for significance performed using Wilcoxon statistics. **c**, Bayesian estimation of the TMRCA using a dataset comprising SNPs in regions of chromosomes 1, 2 and 5 that were co-inherited by all PoT/PoL1 strains. Estimated divergence dates are shown adjacent to key nodes (values inside parentheses indicate 95% CI). Black circles mark nodes that separate different chromosomal haplotypes. Grey shading emphasizes the narrow time window encompassing the predicted divergences of most haplotypes. TMRCA values were used as proxies for the dates of admixes 1 and 2 (Fig. 6).

activity is estimated to have been about 6 years before the first WB outbreak (Fig. 6). All splits occurring outside of this window appear to be artefacts caused by recombination because they involved haplotypes that shared rare recombinant SNPs. Those with later split dates showed recombination between SNPs absent in the PoT1 founding haplotype and must, therefore, have arisen after admix 2. Those showing earlier splits shared SNPs with strain ATCC64557. These variants are unlikely to have been present in the PoL1-1 founder because they were not found in any PoT haplotypes and were therefore probably acquired from a PoL1-1 descendent in the previously inferred backcrosses.

### Host jumping by re-assortment of host-specificity genes

Prior work has shown that *P. oryzae*'s host jump to wheat is correlated with mutation and/or loss of the *PWT3* and *PWT6* host-specificity genes, whose products trigger resistance in wheat[32,33,37]. To understand how the swarm activity might have driven the host jump, we used BLAST searches to trace *PWT3* and *PWT6* alleles through the WB pedigree. Importantly, we found that WB/GLS contained null/non-functional alleles of these genes that already existed as standing variation within *P. oryzae* and the chromosomal patterns of inheritance seen in Fig. 4 were consistent with a scenario where these alleles were donated to the swarm. *PWT6* resides on chromosome 5 in PoE1, but the PoL1-1 founder

inherited the relevant chromosome segment from the PoU1 lineage whose members lack the gene (Extended Data Table 1b). *pwt6^{null}* was then passed down to other PoT/PoL1 members, while others inherited a second null locus from the PoSt donor. The PoL1-1 founder possesses functional *PWT3^A* that resides on a chromosome segment inherited from PoU1 (Fig. 4). *PWT3^A* was passed down to PoT1 in admix 2, and its descendants then acquired non-functional *pwt3* alleles in subsequent matings, with *pwt3^B* having been donated by the PoSt lineage and the transposon-disrupted variant, *pwt^{Atc}*, having been contributed by 2PoE1 (Extended Data Table 1b). Two other non-functional *pwt3* alleles have transposon insertions in the promoter region of genes that are otherwise identical in sequence to *PWT3*:* Neither the MGL retrotransposon[38] in *pwt3^{Atm}* nor the Pot3 insertion in *pwt3^{Atp}* (ref. 39) are present in the PoT1 founder isolates and, therefore, with no evidence of a secondary donation from PoU1, both must have arisen de novo. *pwt3^{Atm}* is found in multiple haplotypic backgrounds consistent with segregation in the swarm, while *pwt3^{Atp}* was found in just a single isolate, WBKY11.

### Discussion

Prior studies implicated a possible role for gene flow in the differentiation of wheat and *Lolium*-infecting lineages of *P. oryzae*[23]. Here, we provide evidence they co-evolved in a multi-hybrid swarm that

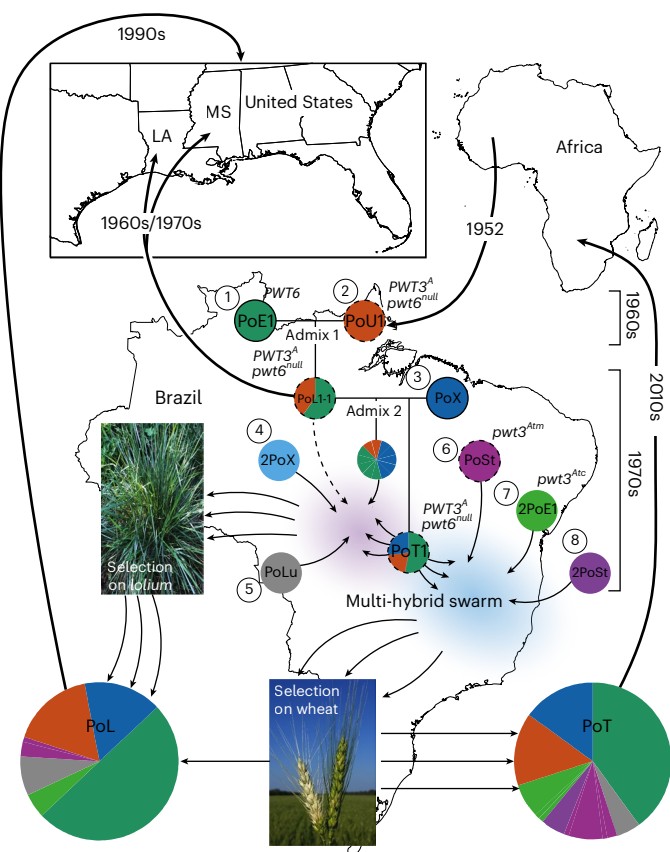

**Fig. 6 | Multi-hybrid swarm model for WB/GLS evolution.** The seven fungal individuals hypothesized to be the main contributors of DNA to PoT/PoL1 are shown as coloured discs with solid outlines depicting the MAT1-1 mating type; dotted outline indicates MAT1-2 and no outline indicates unknown/mixed mating type. PWT alleles, where known, are also shown. Approximate time frames for admixture events are shown and were inferred by molecular dating (Fig. 5). Thin arrows show gene flow and those with dotted lines indicate backcrosses. Thick arrows show suspected migration/importation events. Maps were obtained from Vemaps.com.

via vegetative fusion have also been linked to host range expansions in pathogens such as *Verticillium*[45] and in mutualistic *Epichloë* symbionts[46]. With the resolution afforded by phylogenomic approaches, hybridization between divergent lineages within a single species (admixture) has recently been identified as a potential driver of pathogen diversification and new disease emergence[9,47–49]. For most pathosystems, however, the genetic barriers that prevent colonization by non-pathogens and the mechanism(s) by which hybridization breaks these down are unknown.

The evolutionary history of WB yields new insights in this regard by building on foundational work from Tosa and colleagues who identified five PWT genes that determine *P. oryzae* avirulence on wheat[33,37,50]. After surveying allele distributions for two cloned genes, they proposed an evolutionary model where functional losses of *PWT6* and currently unknown PWT genes allowed an *Eleusine* pathogen to colonize *rwt3* wheat, probably via an intermediate *Lolium* host[32]. A population increase on wheat was then proposed to have been a springboard for the evolution of *pwt3* mutants with virulence on *RWT3* cultivars[33]. By tracing *PWT3/6* alleles through the PoT/PoL1 pedigree (Fig. 6), we found that most of the necessary losses in PWT function occurred via re-assortment of pre-existing virulence alleles that were donated to the swarm by pathogens found on wild and cultivated grasses known to be present in the surrounding agricultural landscape. Loss of *PWT6* occurred in admix 1 but was probably insufficient to establish infectiousness to wheat because all isolates from wheat contain donations from PoX that arrived with admix 2. Presumably, at that time, the PoT1 founder inherited null/non-functional alleles of one or more of the *PWT* genes that remain to be cloned. Virulence to *RWT3* wheat was subsequently gained through donations of *pwt3^b* and *pwt3^{Atc}* from PoSt and 2PoE1, respectively (Fig. 6). Critically, all virulence alleles, including *pwt3^{Atm}* which harbours a de novo transposon insertion, were found in multiple haplotypic backgrounds and, because there was no evidence of genetic exchange having occurred in the epidemic population (Supplementary Data File 2), all of the key losses in *pwt* function appear to have been 'pre-adaptations' that arose within the swarm and, quite possibly, before the wheat host was ever encountered.

Resurrection of ancestral virulence alleles has been proposed as a mechanism by which the apple scab pathogen, *Venturia inequalis*, defeated *Rvi6* resistance in cultivated apples. However, while *avrRvi6* virulence appears to have arisen in populations associated with wild crabapples containing *Rvi6* resistance[51,52], *pwt* alleles in *P. oryzae* came from populations that are specialized on other grasses[24] and have no known history of interaction with wheat. Possibly, the wheat resistance genes that recognize the various PWT avirulence factors have functional equivalents in other Gramineae, similar to the situation with Brassicaceae[53]. In this case, selection for virulence in the canonical grass host might simultaneously defeat a functionally equivalent gene in wheat—or any other grass for that matter. This latter point is especially salient and may explain why so many PoT/PoL1 members (18%) were capable of infecting non-wheat/*Lolium* hosts. Moreover, with some haplotypes being found on multiple genera, and one (PoL1-9) that so far has only been found on oat, it appears that WB/GLS evolution occurred within the context of a broader adaptive radiation that also produced some haplotypes with unusually wide host ranges, and others that are preferentially adapted to other grasses. Interestingly, the fungal pathogens *Zymoseptoria pseudotritici* and *Blumeria graminis triticale* also have recent evolutionary origins in hybrid swarms and high haplotype diversity within the emergent populations furnished to them with expanded host ranges, compared with related species (*Z. pseudotritici*), or their immediate forebears (*B. g. triticale*).

In hindsight it is understandable that shuffling of *pwt* alleles could have facilitated host jumps onto wheat and other grasses, but it is not clear how strains of *P. oryzae* adapted to so many different hosts

repartitioned standing variation present in at least eight individuals from five *P. oryzae* populations individually specialized on *Eleusine*, *Luziola*, *Stenotaphrum*, *Urochloa* and an unknown host related to *Oryza/Setaria* pathogens (Fig. 6). Brazil appears to be the primary centre of origin for both diseases because most haplotypes constituting WB/GLS were not found elsewhere until 1990 and remained restricted to South America until 2016. Very likely, the PoL1-1 founder originated there too because *Eleusine*- and *Urochloa*-infecting donor populations are not found in North America but are endemic and widespread in Brazil's wheat-growing regions[24,40], and the PoE1 segments in its genome are most similar to sequences found in South American *Eleusine* pathogens (Extended Data Fig. 6). It was possibly introduced into the United States via infested hay/seed or by Hurricane Edith, which swept in from central America just 3 months before the 1971 disease outbreaks (Fig. 6). *Urochloa* was first introduced into Brazil in 1952 as a forage to support the beef industry[41], and we suspect that coincident introduction of the *Urochloa*-adapted lineage may have been the initial catalyst for the events whose outcome now threatens global wheat production.

The idea that hybridization drives the evolution of plant pathogens is well established. Interspecific hybrids have been associated with the emergences of several new plant diseases[42,43] and resurgences of old ones (for example, Dutch elm[44]). Allopolyploid hybrids formed

could have come together in the first place, or how progeny of differentially adapted parents avoided epistatic interactions that typically cause losses of virulence on either parental host[42,43]. One possibility is that suitably adapted strains suppress host immunity, allowing co-colonization by less-adapted relatives—a demonstrated avenue for gene flow among different races of the obligate oomycete pathogen, *Albugo candida*[9]. However, recombination in *Albugo* occurred over a considerable period of time and, given the short-lived nature of the WB/GLS swarm and the widespread sharing of recombination blocks (Fig. 4a), we favour a model where all matings from admix 2 onwards occurred contemporaneously in hay containing infected pasture grasses (*Lolium*/*Urochloa*) and associated weeds (*Eleusine*/*Luziola*). Plant tissues in this form would have similarly compromised, or inactive, immunity and should, therefore, have been minimally restrictive to fungal mating and growth.

It has been proposed that host jumping is a 'cornerstone in the evolution of plant pathogens, as it leads to pathogen diversification', which is then 'followed by radiation, specialization and speciation'[4]. The exact opposite appears to be true for WB/GLS because diversification was driven primarily by the recombination of admixture variation, which then led to the host jump, and it would take an estimated 25,000 years to accumulate diversity equivalent to that generated by the brief swarm activity. Moreover, because so few of the mutations that arose after admix 1 were shared among different haplotypes (Supplementary Data File 2), this points to a scenario where most, if not all, of the recombination happened before there was any propagation in the new hosts. As such, we envisage a literal swarm where unfettered mating occurred in a confined area, over a limited period, and generated an untested collection of what might best be described as 'hopeful monsters' (Goldschmidt[44,45]), which only experienced host-driven selection after asexual spores escaped the swarm.

The notion that host jumps drive pathogen diversification comes from an assumption that pathogenic interactions tend to be suboptimal after jumping hosts and that new beneficial mutations must then occur for the reproductive rate to reach a threshold for disease outbreak and epidemic spread ($R_0 > 1$)—a process known as 'fine-tuning'[1–3]. For WB/GLS, the diseases exploded onto the scene so soon after the swarm's formation, it seems unlikely that much fine-tuning would have occurred, especially considering the limited number of new mutations to fuel adaptation. This leads us to propose that the repartitioning of standing variation not only drove the initial host jumps onto wheat and *Lolium* (and other grasses), but produced a recombinant population with individuals sufficiently 'pre-adapted' to the new hosts that they were inherently highly pathogenic. This is unexpected considering the potential for variable epistatic interactions among loci donated by individuals adapted to divergent host species (Fig. 4b). Nevertheless, the number of distinct haplotypes within the outbreak population, and the evenness in their representation, suggests that recombinational load had negligible impact on compatibility. In fact, the hybrids possibly had increased pathogenicity, as has been reported for hybrids of the Dutch elm pathogens, *Ophiostoma ulmi* and *O. novo ulmi*[44], and the anther smut pathogen, *Microbotryum violaceum*[54].

While we do not believe that new mutation played a major role in *P. oryzae*'s establishment on wheat, there has clearly been subsequent adaptation via selection on acquired standing variation. *pwt3^Atc* inherited from the 2PoE1 donor was present at a low frequency pre-1992 but its presence in all isolates collected since 2005 suggests it has been swept to near fixation (Supplementary Data File 1c), presumably in response to widespread cultivation of *RWT3* wheat. This sweep is remarkable because it favoured *pwt3^Atc* alleles in multiple haplotypic backgrounds that show little or no evidence of gene flow between them, yet over the same period other *pwt3* alleles that dominated in the early outbreak population dramatically decreased in frequency. Presumably, *pwt3^Atc* was favoured because the large compound transposon in its open reading frame[26] destroys function, while other alleles, with only SNP variation and/or transposon disruption of their promoters, possibly retain residual activity[33].

Only a few newly emerged fungal plant pathogens have been studied at the phylogenomic level, yet recombination in hybrid swarms is already a recurrent evolutionary theme. *B. g. triticale* and *Z. pseudotritici* also exhibit high haplotype diversity by virtue of their swarm origins but, at the same time, they show almost no variation across chromosome regions inherited from the same parent[8,47]. Thus, they are further examples of new hybrids that, despite potentially confounding effects from recombinational load, became fully ensconced in their respective hosts not long after their evolutionary foundations. Clearly, recycling of existing variation provides for more rapid evolution than a dependency on new beneficial mutations, and admixture/hybridization has the benefit of bringing together divergent sets of alleles that have already been tried-and-tested in related environments[55]. What is most surprising, however, is that despite prior long-term specialization on other host genera, the specific ancestries of large portions of the fungal genomes seemingly have no effective bearing on the immediate compatibility with the new hosts. This suggests that, beyond the primary layer of recognition-based immunity, plant genetic barriers to host jumping by non-pathogens may be less robust than is currently assumed. Indeed, for WB/GLS it appears that the crux of its initial host jump to wheat was simply assembling the right gene combinations to defeat primary recognition. Here, it should be noted that *P. oryzae*, *B. g. triticale* and *Z. pseudotritici* are all pathogens of monocots, so it remains to be seen whether fungal admixture/hybridization can drive similarly rapid host jumps on dicot hosts, or on wild plant species with broader genetic bases. Nevertheless, the surprising story of WB/GLS evolution emphasizes the potentially precarious nature of non-host resistance, especially in the face of new opportunities for hybridization brought about by the anthropogenic movement of fungi.

## Methods

### Fungal cultures

The origins of fungal strains used in this study are listed in Supplementary Data File 1a. Especially important for our study are previously uncharacterized early strains: ATCC64557, which was isolated in the United States in 1980 and immediately deposited in the American Type Culture Collection, and *Triticum* strains collected in Brazil between 1985 and 1989 and maintained desiccated in long-term frozen storage in the Barbara Valent/Forrest Chumley laboratory at the DuPont Company. In 2001, this early *Triticum*-strain collection was moved and permanently stored in the biosafety level 3 *M. oryzae* strain collection maintained at the United States Department of Agriculture Agricultural Research Service Foreign Disease and Weed Science Research Unit in Fort Detrick, Maryland. WB isolates were cultured either under biosafety level 3 containment at Fort Detrick, or under normal laboratory conditions at Universidade Federal de Viçosa, which is in a WB endemic region. Each strain analysed was a fully independent isolate, from a different plant and usually from a different location and/or year. For DNA isolation, all strains were first genetically purified via single-spore isolation and then cultured in 10 ml liquid complete medium (6 g casamino acids, 6 g yeast extract and 10 g sucrose) for 7 days with shaking.

### Next-generation sequencing

DNA was prepared from freeze-dried mycelium using previously described methods[56]. Most libraries were prepared using the Nextera kit (Illumina, cat. no. FC-121-1031) and were constructed according to the manufacturer's instructions, but using a tagmentation step of 60 min. A small number of libraries were prepared using the Kapa HyperPlus kit (Roche, cat. no. 7962401001), according to the manufacturer's protocol. Sequences were acquired using MiSeq (at the University of Kentucky Advanced Genetic Technologies Center, or Bluegrass Community and Technical College) and using HiSeq (150 PE) at Novogene Corp Inc.

## Sequence assembly

Raw reads were trimmed to remove adaptor sequences and poor-quality regions using Trimmomatic (PE mode: ILLUMINACLIP: NexteraPE-PE. fa:2:30:10 SLIDINGWINDOW:20:20 MINLEN:80). Paired-end reads passing these filters were assembled with Newbler 2.9 or Velvet 1.2.10 using VelvetOptimiser to optimize the assembly over the $k$-mer range 59–129.

## SNP calling

We employed a custom perl script[57], which performs two passes of a repeat-masking algorithm originally developed for TruMatch[58] to ensure that SNPs are not called in regions affected by repeat-induced point mutation. Briefly, repeat-masked genomes are aligned using blast (-evalue 1e-20, -max_target_seqs 20000 -outfmt '6 qseqid sseqid qstart qend sstart send btop'), a second repeat-filtering step is then performed, and SNPs are then called only in uniquely aligned regions. iSNPcaller accuracy was tested by identifying SNPs in fully independent genome assemblies derived from the same raw read datasets. This revealed a median false SNP call error rate of -1.1 × 10$^{-5}$ (Farman et al. 2023). The final dataset comprised 471,202 SNPs in unique chromosome regions.

## Phylogenetic and divergence analyses

Pairwise distance data from iSNPcaller were used to build neighbour-joining trees using default parameters in MEGA X (ref. [59]). Tree plots were generated using the radial output format in MEGA X, circular format in ggtree[60] and neighbour-net network in Splitstree5 (ref. [61]). Phylogenetic analyses of *MPG1* and CH7BAC7 were performed by aligning the sequences with MUSCLE 3.8.31 (ref. [62]) and generating maximum likelihood trees with RAXML8 (ref. [63]). A co-phylogeny plot was created in R using the phytools cophylo function with parameters: assoc = NULL, rotate = TRUE[64]. Within-population divergence values were calculated in MEGA X after partitioning the pairwise distance data based on population memberships. Resolution of distinct donor haplotypes from the same source population was accomplished by using the binary haplotype data to build phylogenetic trees in RAxML under the bingamma substitution model.

## Determination of population subdivision

SNP data from the iSNPcaller output files were downsampled by retaining every tenth SNP along each chromosome and converted into STRUCTURE format using a custom script. The find.clusters function in the Bayesian information criterion/discriminant analysis of principal components (DAPC) module from Poppr[65] was used to identify clusters (max.n.clust = 30, n.iter = 1 × 10$^6$). DAPC was performed over a $K$-value range from 10 to 25, after retaining 100 principal components. Populations used for subsequent analyses were defined according to the partitioning at $K$ = 16, as this value had the lowest Bayesian information criterion value. Similar population divisions were identified with STRUCTURE[66], although the latter programme failed to resolve the PoO and PoS lineages, which were clearly separated, both by DAPC (Extended Data Fig. 3) and by phylogenetic approaches[23]. Lineages were named with a Po prefix, with suffixes corresponding to the primary host genus and a numerical identifier if the genus harboured more than one population.

## Assessing chromosome ancestry using ChromoPainter

First, we used a custom script to call chromosomal haplotypes (only bi-allelic sites) for each strain, based on the SNP data from iSNPcaller. The dataset was filtered to remove all SNP positions where any one strain had a missing data point. ChromoPainter was run separately for a single representative of each PoT/PoL1 haplotype, using all non-PoT/PoL1 strains as potential donors, and an even recombination rate of 7 × 10$^{-9}$ across all chromosomes. Run parameters included the haploid (-j) and print copyprobsperlocus files (-b) options, with each expectation maximization being performed over ten iterations,

while maximizing over copy proportions (-ip option). Donor probability plots and chromosome haplotype plots were generated from the resulting copyprobs files, using custom R programs. To speed up plotting and generate plots with manageable sizes, the copyprobs file can be downsampled by rounding chromosome coordinates to the nearest 10 kb and retaining just one data point per interval. The inferred donor contributions were obtained by using the write.csv() function to print out the data frame used for the ChromoPainter stacked bar chart plots. Overall donor contributions to each haplotype/population were obtained by parsing the ChromoPainter chunklengths.out file, using custom R scripts.

## Assessing chromosome ancestry using ShinyHaplotypes

The ChromoPainter input files were used as inputs to the perl script SlideCompare.pl, which compares haplotypes in a pairwise fashion in sliding windows of 2,000 variant sites and a step size of 400. Haplotype divergence was calculated as number of SNPs per variant site. The output files were then read by the ShinyHaplotypes.R code for interactive and static plotting.

## Reciprocal crossover analyses

A custom perl script was used to interrogate the data frame used for the ChromoPainter plots. Briefly, the script identifies chromosome positions showing reciprocal swaps in parentage across the same inter-SNP interval. It then reports the positions of the crossover, which haplotypes exhibit the crossover and the respective heritages of the chromosome segments on each side of the exchange.

## Tip dating using BEAST 2.6.3

For this analysis, we used only sequences where all strains were predicted to have inherited the respective chromosome regions from the original founder isolate (Chr1: 2,300,000–3,400,000, Chr2: 500,000–1,500,000 and Chr5: 300,000–2,000,000). By restricting our analysis to sequence data from these regions we could be reasonably confident that the substitution rate was inferred using only new variants that arose between the founder event and each strain's collection date, thereby avoiding inflation due to pre-existing SNPs acquired by admixture. In addition, because we suspected that the new populations had very recent evolutionary origins, we sought to minimize 'noise' from false SNP calls by using a dataset that represents the intersection of calls generated using two completely different SNP calling methods, both of which were pre-filtered to avoid false calls in repeated sequences. In addition to using the data from iSNPcaller, we aligned raw read data to the B71 reference genome using bowtie and then recalled SNPs in the target regions using the Genome Analysis Toolkit with the parameters: HaplotypeCaller, -ploidy 1; -emit-ref-confidence Genomic variant call format (VCF); GenotypeGenomicVCFs default settings; and SelectVariants -select-type SNP. A custom script (Smart-SNPs.pl[67]) was used to filter the resulting VCF files using the following schema: SNPs should not be repeated regions of the B71 reference, SNPs should not be heterozygous (*P. oryzae* is haploid so heterozygous calls indicate the calling of false SNPs that reside in repeated regions of the query genome), the alternate:reference allele ratio must be ≥20:1, and there must be a minimum of ten reads covering the site in question. SNPs passing these pre-qualifiers were then cross-referenced with the variants identified by iSNPcaller and inconsistencies were resolved by interrogating the raw read data to determine true presence/absence of the SNP.

The strength of the phylogenetic signal in the dataset was assessed by using RAxML to build a maximum likelihood tree under the GTRgamma model, with 1,000 bootstrap replications. Patristic distances to ATCC64557 (the earliest sampled isolate) were then calculated using the cophenetic function in ape[68], and the correlation with sampling date was determined using the Pearson method. The robustness of the signal was assessed by comparing the distribution of

Pearson's *r* after resampling the dataset with replacement (*n* = 1,000), versus randomizing the sampling dates (*n* = 1,000). The phylogenetic signal operations were performed using a custom script modified from[69].

For molecular dating, the dataset was analysed as a single partition due to the small number of variant sites. The HKY + I + G site substitution model was utilized as determined to be optimal by PartitionFinder2 (ref. [70]). Constant sites were determined by using a custom perl script to count the invariant nucleotides in the queried portions (that is, non-repeated) portions of the surveyed chromosomes. Both strict clock and relaxed log normal clock models were employed, and their rates were estimated in separate Markov chain Monte Carlo runs. Coalescent Extended Bayesian Skyline was selected as the tree prior and 20 independent Markov chain Monte Carlo samplings were performed for each parameter set, with each incorporating 10 million pre-burnin iterations, 100 million sampled iterations and data logging every 5,000 iterations. The resulting logfiles were inspected in Tracer v1.7.1 and the five with the highest effective sample size values were combined using LogCombiner, along with the corresponding tree files[71]. TreeAnnotator was used to create a maximum clade credibility tree which was plotted using the ape and phyloch packages for R (ref. [72]).

### Analysis of PWT gene alleles
The *PWT3* and *PWT6* gene sequences were used to search genome assemblies for all available strains using BLAST and allele assignments were made based on nucleotide mismatches and gaps. For transposon-disrupted alleles, the positions of transposon insertions and target site duplications were used to define alleles. *pwt6^null* alleles were characterized by first aligning the B71 chromosomes with the *PWT6* locus in the PoE1 isolate, CD156, to identify the deletion breakpoints and then by comparing the structure of the breakpoint and flanking sequence between PoT/PoL1 members and isolates from the inferred donor populations. Finally, we verified that the allele designation for a given isolate was consistent with the inferred heritage of the corresponding chromosome segment, as determined by chromopainting.

### Reporting summary
Further information on research design is available in the Nature Portfolio Reporting Summary linked to this article.

## Data availability
Sequence data are available under various NCBI accessions as listed in Supplementary Data File 1a. Datasets used for the analyses described herein are available at https://github.com/drdna/WheatBlastEvolution.

## Code availability
Custom bash, perl and R codes used to perform the analyses and generate figures are available on GitHub (https://github.com/drdna/WheatBlastEvolution).

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

## Acknowledgements

We thank Y. Tosa for sharing unpublished genome sequence data, and the following for helpful comments: S. Kamoun and T. Langner (the Sainsbury Laboratory), P. Gladieux (Institut National de la Recherche Agronomique), N. Grunwald (Oregon State University), D. Weisrock (University of Kentucky) and L. Trevathan (Animal and Plant Health Inspection Service). We thank the University of Kentucky Center for Computational Sciences and Information Technology Services Research Computing for their support and use of the Morgan and Lipscomb Compute Clusters and associated resources. This work was supported by the United States Department of Agriculture, Agriculture and Food Research Initiative grants 2013-68004-20378 and 2021-68013-33719 (B.V.), multistate project NE1602 (M.F.); Agricultural Research Service project 8044-22000-046-00D (B.V.); Hatch project KY012037 (M.F.); the National Science Foundation, MCB-1716491 (M.F.); and the University of Kentucky College of Agriculture Food and the Environment (M.F.). This is contribution no. 21-121-J from the Kansas Agricultural Experiment Station. Mention of trade names or commercial products in this publication is solely for the purpose of providing specific information and don't imply recommendation or endorsement by the U.S. Department of Agriculture. USDA is an equal opportunity provider and employer.

## Author contributions

M.R., B.C., J.R.D., E.M.D.P., B.V. and M.L.F. were involved in study design. J.P.A., K.F.P., S.M. and B.V. performed sample collection, strain and DNA isolation. M.R., B.C. and M.L.F. performed the bioinformatic analyses and created custom script. M.L.F. and M.R. wrote the manuscript, deposited data at NCBI and created the GitHub repository. All authors edited the manuscript and approved the final version.

## Competing interests

The authors declare no competing interests.

## Additional information

**Extended data** is available for this paper at https://doi.org/10.1038/s41559-023-02237-z.

**Correspondence and requests for materials** should be addressed to Mark L. Farman.

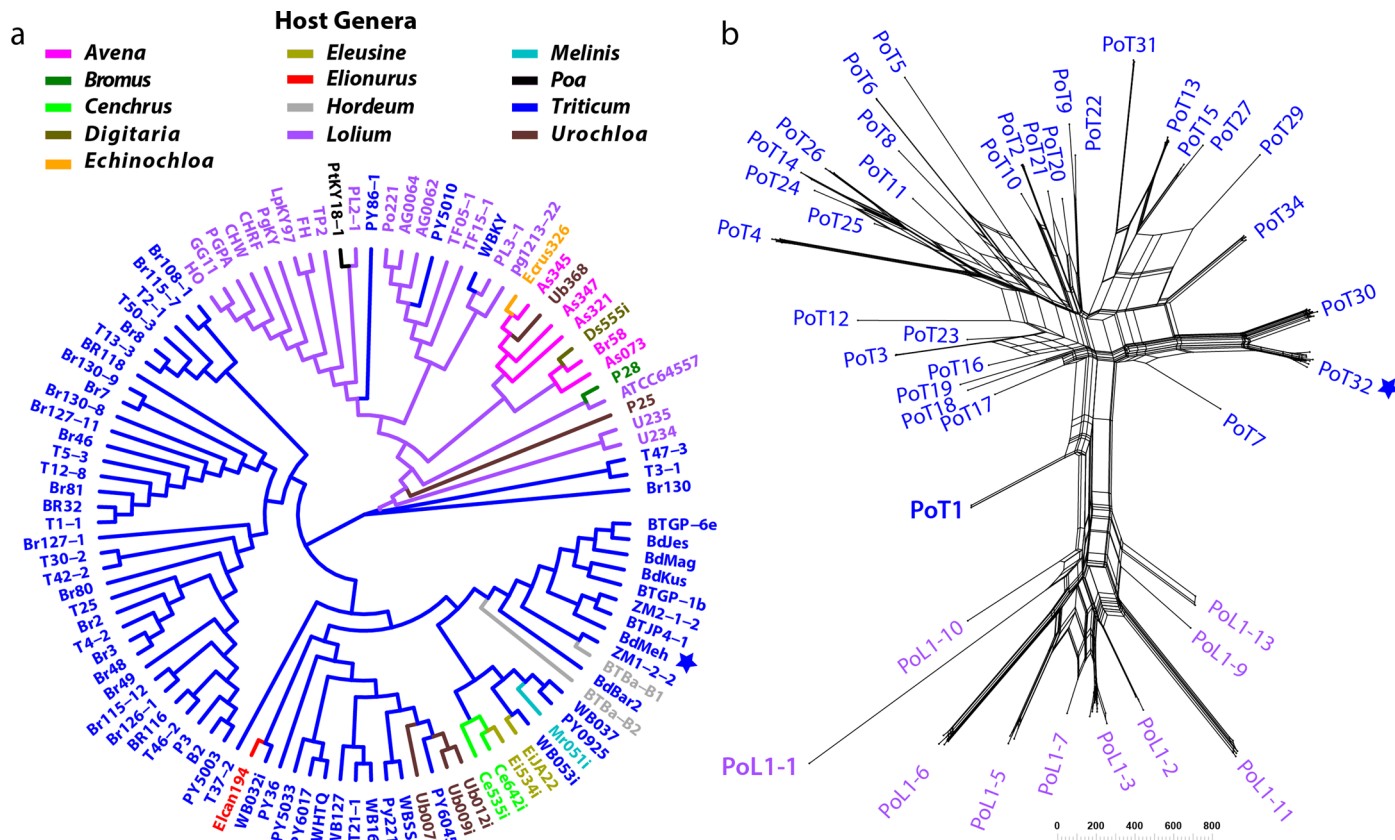

**Extended Data Fig. 1 | Phylogenetic analyses of the PoT/PoL1 lineages.** The tree and the neighbor-network were constructed using whole genome pairwise distance data for isolates that grouped within PoT/PoL1, regardless of the host genus from which they were recovered. **a**) Relaxed host specificity indicated by the recovery of PoT/PoL1 members from 11 additional host genera. Isolate names and branches are colored according to the host-of-origin. The tree was drawn using ggtree with the branch.length = 'none' option selected. **b**) Network showing deep reticulations consistent with a history of extensive recombination among member isolates. Network built using built using the NeighborNet algorithm in Splitstree5. Labeled clades/taxons correspond to distinct chromosomal haplotypes and are numbered according to the earliest date a member isolate was sampled. Scales represent SNPs/Mb of uniquely aligned genome sequence. The B71 reference strain was not included in this analysis but shows near sequence identity to isolate BdMeh whose position is highlighted with a blue star.

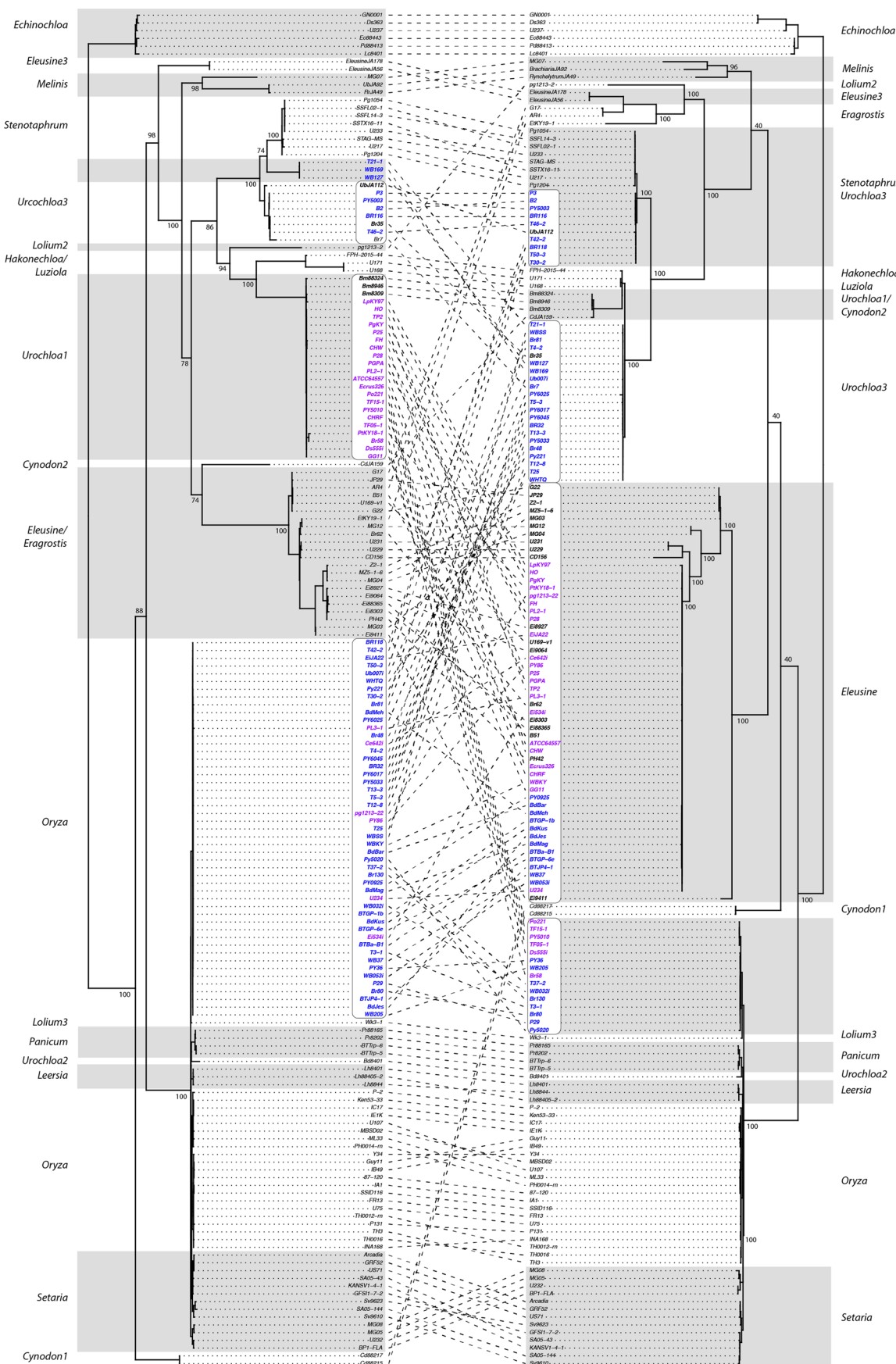

**Extended Data Fig. 2 | Discordant phylogenies for two linked loci on chromosome 7.** Phylogenetic trees were generated for 20 kb of sequence surrounding markers *MPG1* (Chr7:159,716-179,715) (right-hand tree) and CH7BAC7 (Chr7:1,173,687-1,193,686) (left-hand tree) using a GTR gamma model with 100 bootstrap replications. Tips were re-ordered and plotted using the cophylo function in phytools. PoT lineage members are labeled in blue and PoL1 members in purple.

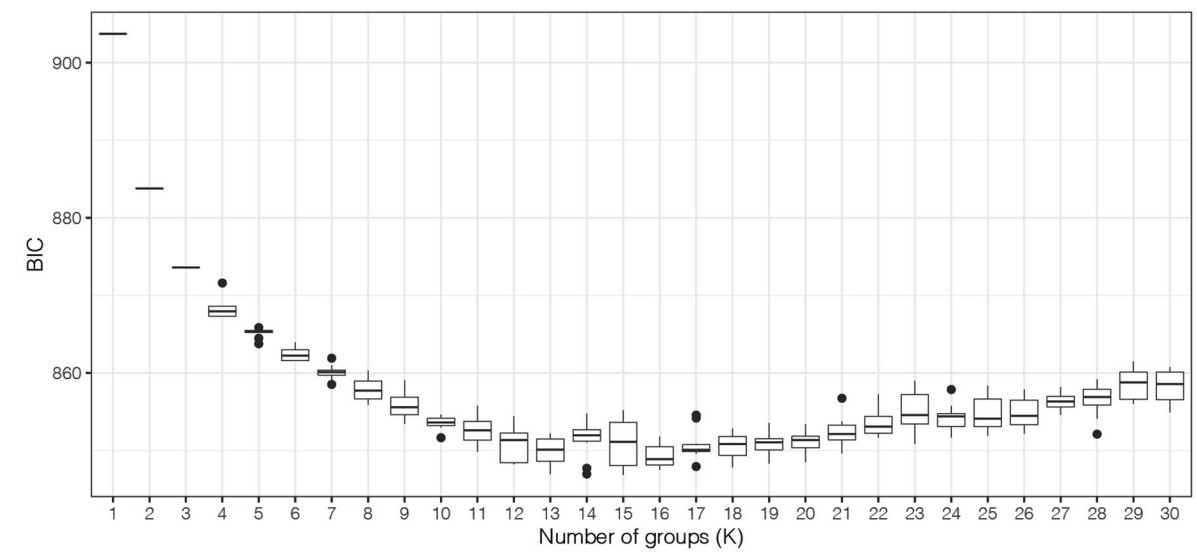

**Extended Data Fig. 3 | Discriminant Analysis of Principle Components (DAPC) to discern *P. oryzae* lineage memberships. a**) Use of Bayesian Information Criterion to determine the most probable number of discrete lineages. Bars = median; box = interquartile range (IQR); whiskers = smallest/largest value within 1.5 times respective IQR limit. The lowest median BIC value was at K = 16. **b**) Assignment of lineage memberships using DAPC. Colors define the lineage membership for each isolate listed at the bottom for K values from 10 to 25.

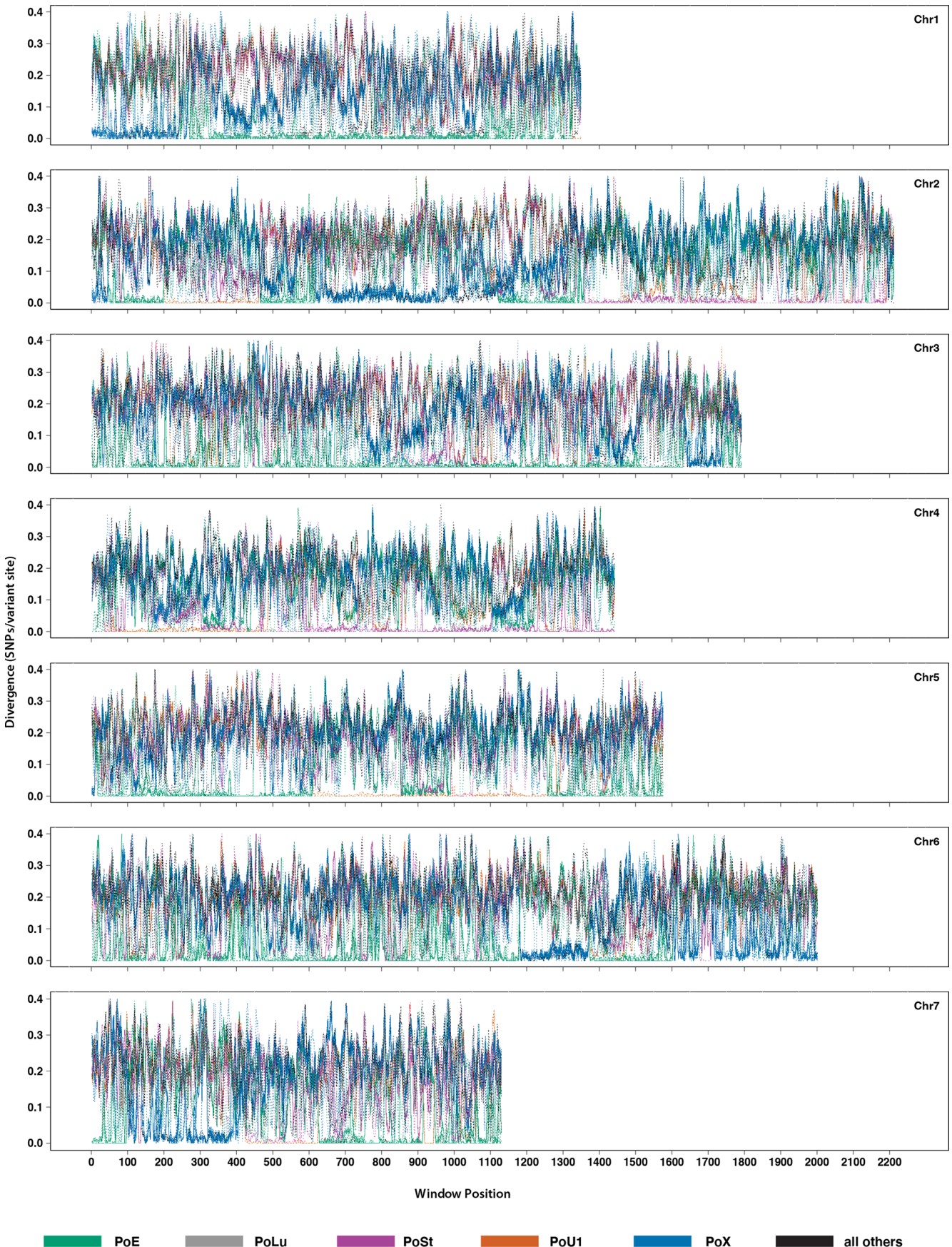

**Extended Data Fig. 4 | Haplotype divergence between B71 and 96 candidate donor isolates.** All 96 plots have been overlaid in the figure and lines are colored according to each isolate's lineage designation. Lineages related to PoX (PoO, PoS, PoLe, and PoP) are colored blue. Dotted lines are used to improve visualization of overlapping plots.

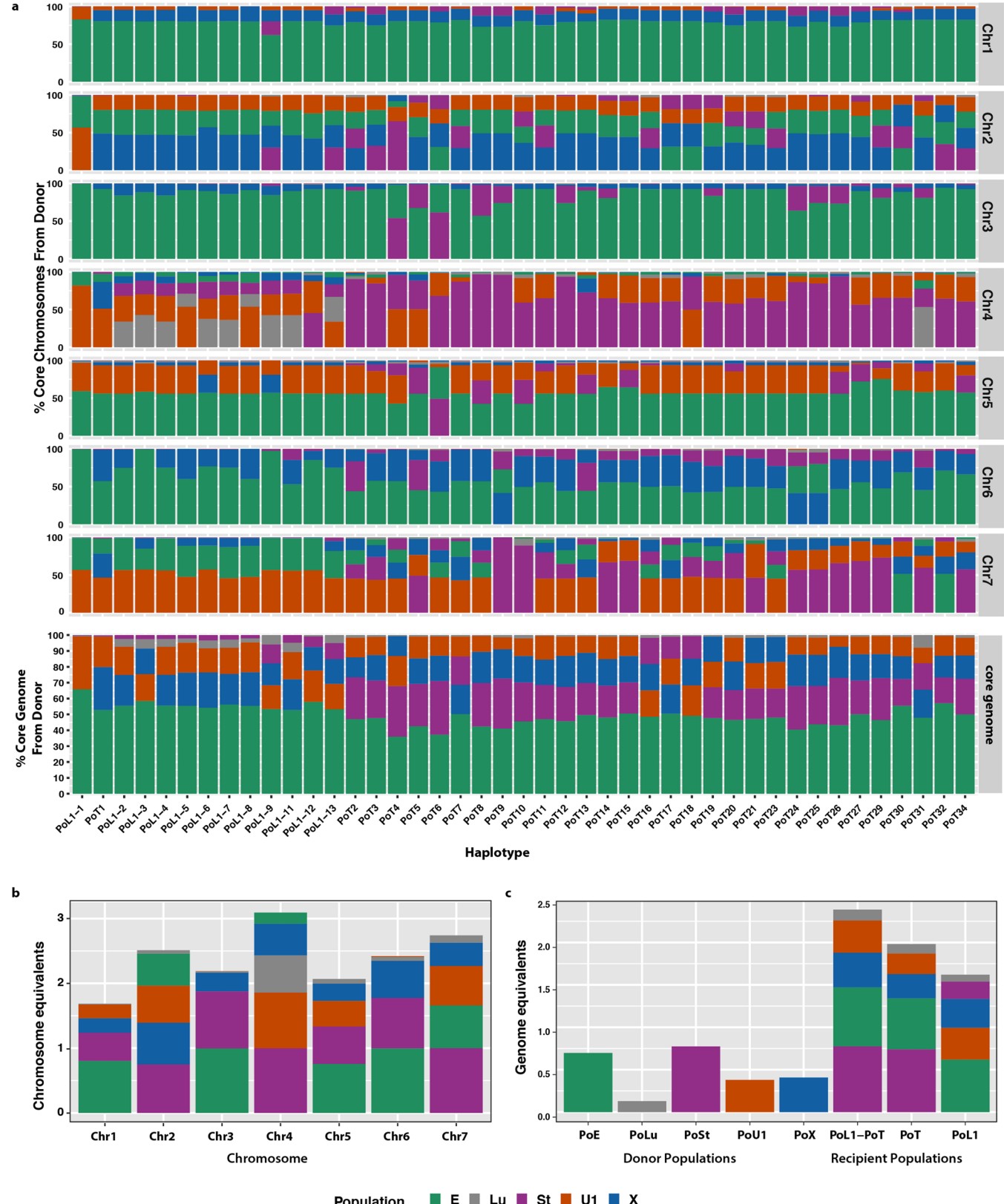

**Extended Data Fig. 5 | Proportions of PoT/PoL1 genomes contributed by the five donor lineages.** a) Relative donor contributions to each chromosome and the genome as a whole. b) Chromosome equivalents contributed by each donor lineage. c) Genome equivalents contributed by each donor lineage, and total genome equivalents present in the recipient lineages (=estimate of pangenome size). In all plots, stacking order is from the greatest contribution to least. Only contributions from the main donors are shown (other contributions were less than 1%).

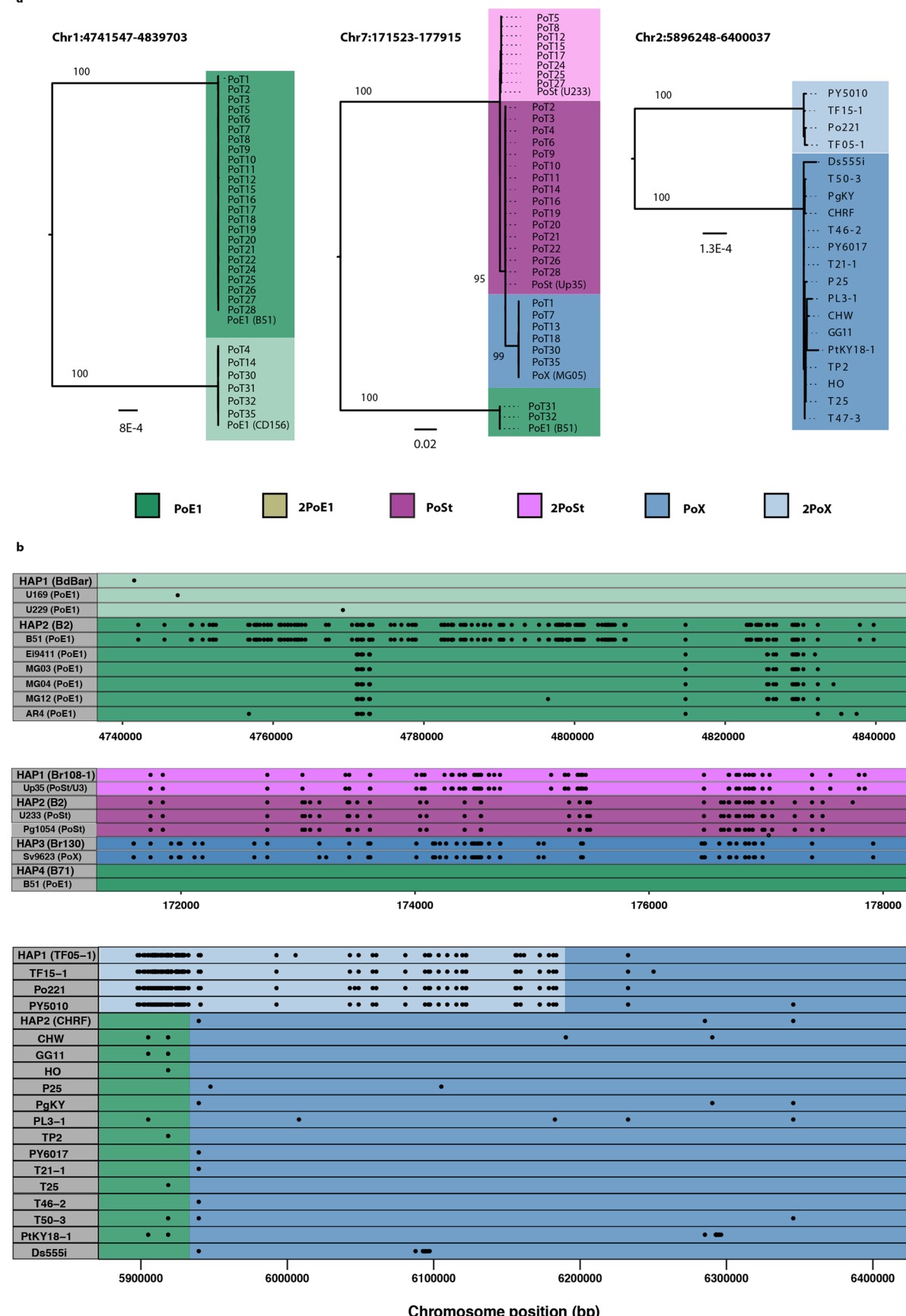

**Extended Data Fig. 6 | See next page for caption.**

**Extended Data Fig. 6 | Identification of secondary introgressions from the PoSt and PoE1 lineages. a**) Maximum likelihood trees built using binary haplotype data for: PoE1 contributions on chromosome 1 between positions 4,741,547 and 4,839,703; PoT isolates and candidate donors on chromosome 7 (171,523 to 177,915); and PoX contributions on chromosome 3 (5896248 to 6400037). The trees were built using the bingamma substitution model and 100 bootstrap replications. Distances represent nucleotide divergence. **b**) Plots showing nucleotide differences for select isolates across the regions used to build the trees in a. The HAP# designation is used to label one example of each distinct haplotype identified across the three regions analyzed. Tracks are colored according to inferred donor (see legend for a).

**Extended Data Table 1 | PoT/PoL1 haplotypes harboring reciprocal crossover products**

| Chromosome | pre-X over SNP pos. | pre-X over allele | post-X-over SNP pos. | post-X-over allele | Haplotype(s) |
|:---:|:---:|:---:|:---:|:---:|:---|
| 1 | 6278974 | E | 6280676 | St | PoT26 |
|  | 6278974 | St | 6280676 | E | PoT2 |
| 3 | 7340282 | St | 7354813 | E | PoL1-6, PoL1-7 |
|  | 7340282 | E | 7354813 | St | PoT14, PoT31, PoT4, PoT30, PoT20 |
| 4 | 164223 | E | 182460 | St | PoT5, PoT22, PoT12 |
|  | 164223 | St | 182460 | E | PoT34 |
|  | 357691 | U1 | 385442 | St | PoT13 |
|  | 357691 | St | 385442 | U1 | PoT31, PoT6 |
|  | 4471461 | Lu | 4573068 | E | PoL1-3, PoL1-4, PoL1-7, PoL1-13 |
|  | 4471461 | E | 4573068 | Lu | PoL1-5, PoL1-8 |
| 6 | 4440024 | X | 4441248 | E | PoT2 |
|  | 4440024 | E | 4441248 | X | PoT18 |
|  | 6021357 | X | 6057434 | St | PoT10, PoT17, PoT21 |
|  | 6021357 | St | 6057434 | X | PoT14, PoT4, PoT30, PoT20, PoT24 |
| 7 | 385442 | St | 525372 | X | PoT9, PoT3 |
|  | 385442 | X | 525372 | St | PoT32, PoT31, PoT2, PoT34, PoT13, PoT18 |
|  | 3517584 | St | 3538663 | E | PoT14, PoT34, PoT30 |

**Extended Data Table 2 | Donor genome proportions inherited by each PoL1/PoT haplotype**

| Haplotype | PoE1+PoE2 | PoLu | PoSt | PoU1 | PoX |
|-----------|-----------|------|------|------|-----|
| PoL1-1 | 65.9 | 0.3 | 0.3 | 33.3 | 0.2 |
| PoL1-2 | 58.5 | 5.9 | 2.6 | 16.9 | 16.0 |
| PoL1-3 | 55.6 | 4.9 | 2.4 | 17.8 | 19.3 |
| PoL1-4 | 55.6 | 4.9 | 2.5 | 17.7 | 19.3 |
| PoL1-5 | 55.4 | 2.7 | 2.2 | 18.9 | 20.8 |
| PoL1-6 | 54.2 | 5.1 | 3.3 | 15.2 | 22.2 |
| PoL1-7 | 56.3 | 5.2 | 2.9 | 16.5 | 19.2 |
| PoL1-8 | 55.5 | 2.6 | 2.1 | 18.9 | 21.0 |
| PoL1-9 | 52.9 | 6.0 | 4.8 | 17.2 | 19.2 |
| PoL1-10 | 57.9 | 0.9 | 6.7 | 19.9 | 14.6 |
| PoL1-11 | 53.4 | 6.0 | 11.9 | 15.0 | 13.7 |
| PoL1-13 | 53.2 | 5.1 | 10.1 | 16.1 | 15.5 |
| PoT1 | 52.8 | 0.4 | 0.7 | 19.2 | 26.9 |
| PoT2 | 47.8 | 1.2 | 23.7 | 11.4 | 15.9 |
| PoT3 | 36.0 | 0.7 | 31.9 | 18.9 | 12.5 |
| PoT4 | 48.0 | 7.9 | 16.8 | 9.7 | 17.6 |
| PoT5 | 47.1 | 1.7 | 26.3 | 12.4 | 12.5 |
| PoT6 | 37.3 | 0.9 | 33.8 | 12.1 | 15.9 |
| PoT7 | 50.2 | 1.0 | 18.0 | 12.4 | 18.3 |
| PoT8 | 42.4 | 0.6 | 27.4 | 10.0 | 19.5 |
| PoT9 | 42.6 | 1.0 | 26.7 | 13.8 | 16.0 |
| PoT10 | 45.6 | 2.1 | 24.6 | 11.2 | 16.5 |
| PoT11 | 47.0 | 0.6 | 21.6 | 14.9 | 15.8 |
| PoT12 | 45.7 | 1.1 | 21.7 | 12.1 | 19.4 |
| PoT13 | 49.6 | 0.8 | 20.2 | 10.5 | 18.9 |
| PoT14 | 48.1 | 1.2 | 20.0 | 14.1 | 16.6 |
| PoT15 | 41.1 | 1.4 | 31.5 | 7.5 | 18.4 |
| PoT16 | 50.6 | 0.9 | 19.7 | 12.3 | 16.5 |
| PoT17 | 48.6 | 1.8 | 16.5 | 16.6 | 16.5 |
| PoT18 | 50.5 | 1.1 | 13.8 | 16.3 | 18.2 |
| PoT19 | 49.1 | 0.7 | 14.2 | 19.2 | 16.8 |
| PoT20 | 47.7 | 1.0 | 19.4 | 16.1 | 15.8 |
| PoT21 | 46.5 | 1.6 | 18.8 | 15.1 | 18.0 |
| PoT22 | 47.2 | 1.8 | 19.1 | 16.0 | 15.9 |
| PoT24 | 48.0 | 1.3 | 18.3 | 16.7 | 15.7 |
| PoT25 | 40.5 | 1.3 | 27.4 | 11.0 | 19.8 |
| PoT26 | 43.8 | 1.6 | 24.2 | 10.9 | 19.6 |
| PoT27 | 43.3 | 0.9 | 29.9 | 6.6 | 19.4 |
| PoT28 | 50.1 | 1.1 | 21.3 | 11.0 | 16.4 |
| PoT30 | 46.5 | 0.7 | 26.4 | 11.3 | 15.1 |
| PoT31 | 55.5 | 1.2 | 16.8 | 12.2 | 14.2 |
| PoT32 | 57.1 | 0.6 | 16.3 | 12.6 | 13.5 |
| PoT34 | 50.1 | 1.7 | 22.4 | 11.2 | 14.6 |

# Reporting Summary

## Statistics

For all statistical analyses, confirm that the following items are present in the figure legend, table legend, main text, or Methods section.

| n/a | Confirmed | |
|---|---|---|
| ☒ | ☐ | The exact sample size (*n*) for each experimental group/condition, given as a discrete number and unit of measurement |
| ☒ | ☐ | A statement on whether measurements were taken from distinct samples or whether the same sample was measured repeatedly |
| ☒ | ☐ | The statistical test(s) used AND whether they are one- or two-sided<br>*Only common tests should be described solely by name; describe more complex techniques in the Methods section.* |
| ☒ | ☐ | A description of all covariates tested |
| ☒ | ☐ | A description of any assumptions or corrections, such as tests of normality and adjustment for multiple comparisons |
| ☒ | ☐ | A full description of the statistical parameters including central tendency (e.g. means) or other basic estimates (e.g. regression coefficient) AND variation (e.g. standard deviation) or associated estimates of uncertainty (e.g. confidence intervals) |
| ☒ | ☐ | For null hypothesis testing, the test statistic (e.g. *F*, *t*, *r*) with confidence intervals, effect sizes, degrees of freedom and *P* value noted<br>*Give P values as exact values whenever suitable.* |
| ☐ | ☒ | For Bayesian analysis, information on the choice of priors and Markov chain Monte Carlo settings |
| ☒ | ☐ | For hierarchical and complex designs, identification of the appropriate level for tests and full reporting of outcomes |
| ☒ | ☐ | Estimates of effect sizes (e.g. Cohen's *d*, Pearson's *r*), indicating how they were calculated |

*Our web collection on statistics for biologists contains articles on many of the points above.*

## Software and code

Policy information about availability of computer code

| | |
|---|---|
| Data collection | no software was used for data collection |
| Data analysis | Trimmomatic 0.39, velvet 1.2.10, Newbler 2.9/3.0, Bowtie2 2.2.5, Genome Analysis Tool Kit 4.1.3.0, picard 2.21.6, ChromoPainterV2,MEGA X, RAxML-NG 0.9.0, Partition finder 2.1.1, BEAST 2.9, ShinyHaplotypes 2, and various custom code (available on GitHub) |

For manuscripts utilizing custom algorithms or software that are central to the research but not yet described in published literature, software must be made available to editors and reviewers. We strongly encourage code deposition in a community repository (e.g. GitHub). See the Nature Portfolio guidelines for submitting code & software for further information.

## Data

Policy information about availability of data

All manuscripts must include a data availability statement. This statement should provide the following information, where applicable:
- Accession codes, unique identifiers, or web links for publicly available datasets
- A description of any restrictions on data availability
- For clinical datasets or third party data, please ensure that the statement adheres to our policy

Data availability
Sequence data are available under various NCBI accessions as listed in
Supplementary Data File 1. Some of the datasets used for the analyses described herein are available at https://github.com/drdna/WheatBlastEvolution. Datasets

that are too large to be uploaded to public repositories can be provided upon request from the communicating author.
Code availability
Custom bash, perl and R codes used to perform the analyses and generate figures are available on GitHub: (https://github.com/drdna/WheatBlastEvolution).

# Research involving human participants, their data, or biological material

Policy information about studies with [human participants or human data](). See also policy information about [sex, gender (identity/presentation), and sexual orientation]() and [race, ethnicity and racism]().

| | |
|---|---|
| Reporting on sex and gender | *Use the terms sex (biological attribute) and gender (shaped by social and cultural circumstances) carefully in order to avoid confusing both terms. Indicate if findings apply to only one sex or gender; describe whether sex and gender were considered in study design; whether sex and/or gender was determined based on self-reporting or assigned and methods used. Provide in the source data disaggregated sex and gender data, where this information has been collected, and if consent has been obtained for sharing of individual-level data; provide overall numbers in this Reporting Summary. Please state if this information has not been collected. Report sex- and gender-based analyses where performed, justify reasons for lack of sex- and gender-based analysis.* |
| Reporting on race, ethnicity, or other socially relevant groupings | *Please specify the socially constructed or socially relevant categorization variable(s) used in your manuscript and explain why they were used. Please note that such variables should not be used as proxies for other socially constructed/relevant variables (for example, race or ethnicity should not be used as a proxy for socioeconomic status). Provide clear definitions of the relevant terms used, how they were provided (by the participants/respondents, the researchers, or third parties), and the method(s) used to classify people into the different categories (e.g. self-report, census or administrative data, social media data, etc.) Please provide details about how you controlled for confounding variables in your analyses.* |
| Population characteristics | *Describe the covariate-relevant population characteristics of the human research participants (e.g. age, genotypic information, past and current diagnosis and treatment categories). If you filled out the behavioural & social sciences study design questions and have nothing to add here, write "See above."* |
| Recruitment | *Describe how participants were recruited. Outline any potential self-selection bias or other biases that may be present and how these are likely to impact results.* |
| Ethics oversight | *Identify the organization(s) that approved the study protocol.* |

Note that full information on the approval of the study protocol must also be provided in the manuscript.

# Field-specific reporting

Please select the one below that is the best fit for your research. If you are not sure, read the appropriate sections before making your selection.

☐ Life sciences  ☐ Behavioural & social sciences  ☒ Ecological, evolutionary & environmental sciences

For a reference copy of the document with all sections, see [nature.com/documents/nr-reporting-summary-flat.pdf]()

# Ecological, evolutionary & environmental sciences study design

All studies must disclose on these points even when the disclosure is negative.

| | |
|---|---|
| Study description | The experiment was designed so as to determine the genetic basis for accumulation of enormous quantities of population variation over an extremely short period of time. We used 90 fungal isolates from the study population and 96 representing other host-specialized forms. We initially used genome alignments to characterize the variation and then various population genomic tools to identify, characterize and quantity genetic admixture events. Finally, we used Bayesian methods to estimate the time scale for the new population's evolution. |
| Research sample | Contemporarily-sampled fungal strains collected from regions exhibited outbreaks of disease caused by Pyricularia species; historical samples from culture collections; genomic data from the global research community - either downloaded from national data repositories, or provided by individuals scientists. |
| Sampling strategy | Contemporary samples and historical were collected from regions exhibiting wheat blast outbreaks; data were gathered from all available sources, with the goal of maximizing representation of Pyricularia isolates from as many host genera/species as possible |
| Data collection | Data were collected automatically by sequencing machines, or downloaded from online repositories. Associated metadata were gathered at the same time. |
| Timing and spatial scale | Data were gathered continuously from 2008 to 2020 |
| Data exclusions | the only data that were excluded were poor quality genome assemblies that yielded an unacceptable high number of false genotyping calls |

| Reproducibility | Each result is supported by multiple, independent lines of evidence. All data, no matter what tools/analytical approaches were fully consistent |
|---|---|
| Randomization | N/A |
| Blinding | N/A |

Did the study involve field work? ☐ Yes ☒ No

# Reporting for specific materials, systems and methods

We require information from authors about some types of materials, experimental systems and methods used in many studies. Here, indicate whether each material, system or method listed is relevant to your study. If you are not sure if a list item applies to your research, read the appropriate section before selecting a response.

## Materials & experimental systems

| n/a | Involved in the study |
|---|---|
| ☒ ☐ | Antibodies |
| ☒ ☐ | Eukaryotic cell lines |
| ☒ ☐ | Palaeontology and archaeology |
| ☒ ☐ | Animals and other organisms |
| ☒ ☐ | Clinical data |
| ☒ ☐ | Dual use research of concern |
| ☒ ☐ | Plants |

## Methods

| n/a | Involved in the study |
|---|---|
| ☒ ☐ | ChIP-seq |
| ☒ ☐ | Flow cytometry |
| ☒ ☐ | MRI-based neuroimaging |

