## [Peer Review File · Nature Ecology & Evolution]

Peer Review Information

Journal: Nature Ecology & Evolution

Manuscript Title: RECENT CO-EVOLUTION OF TWO PANDEMIC PLANT DISEASES IN A MULTI-HYBRID SWARM

Corresponding author name(s): Mark L. Farman

Editorial Notes:

Reviewer Comments & Decisions:

Decision Letter, initial version:

26th November 2021

Dear Professor Farman,

Your manuscript entitled "Recombination of standing variation in a multi-hybrid swarm drove adaptive radiation in a fungal pathogen and gave rise to two pandemic plant diseases" has now been seen by two reviewers, whose comments are attached. The reviewers have raised a number of concerns which will need to be addressed before we can offer publication in Nature Ecology & Evolution. We will therefore need to see your responses to the criticisms raised and to some editorial concerns, along with a revised manuscript, before we can reach a final decision regarding publication.

We therefore invite you to revise your manuscript taking into account all reviewer and editor comments. Please highlight all changes in the manuscript text file in Microsoft Word format.

* If you have not done so already please begin to revise your manuscript so that it conforms to our Article format instructions at <http://www.nature.com/natecolevol/info/final-submission>. Refer also to any guidelines provided in this letter.

2[REDACTED]

Nature Ecology & Evolution is committed to improving transparency in authorship. As part of our efforts in this direction, we are now requesting that all authors identified as 'corresponding author' on published papers create and link their Open Researcher and Contributor Identifier (ORCID) with their account on the Manuscript Tracking System (MTS), prior to acceptance. ORCID helps the scientific community achieve unambiguous attribution of all scholarly contributions. You can create and link your ORCID from the home page of the MTS by clicking on 'Modify my Springer Nature account'. For more information please visit www.springernature.com/orcid.

[REDACTED]

Reviewer expertise:

Reviewer #1: ecology and evolution of fungi, including plant pathogens, phylogenetics

Reviewer #2: genomics, ecology and evolution of pathogenic fungi

Reviewers' comments:

Reviewer #1 (Remarks to the Author):

The manuscript submitted by Rahnama et al. describes the evolution of two new pathogen lineages causing wheat blast and gray leaf spot of ryegrasses by using extensive genomic analyses. Authors convincingly show that the new lineages emerged through several admixture events between divergent host specific pathogen lineages and they state that admixed variation fuelled pathogen adaptive radiation. Unquestionable, the strength of the manuscript is the exceptional genomic dataset

2which was used to underline their hypothesis. The idea to use this dataset to underpin a very recent and widely discussed topic in evolutionary biology is striking. However, the manuscript revealed several major flaws that need to be addressed before publication in nature ecology and evolution. The manuscript is extraordinary difficult to read and understand. This is surely because of the very complex story of pathogen evolution authors want to sell and tell but to a large extent also due to the manuscript structure and the presentation of results. For example, the whole story is built on previous work about parallel evolution and admixture variation. These quite complex concepts and theories are introduced in no more than ten lines at the beginning of the introduction. Words and terms such as admixture variation, hybrid swarm, populations, adaptive radiation are not well introduced, defined and/or explained. The wording is sometimes inappropriate and contains a lot of personal statements and arguing such as "we believe...", "we would simply point out...". The presentation of figures is insufficient and the overall impression about the manuscript is "drafty". The following major comments should help authors to improve the manuscript prior resubmission.

Introduction:

- needs a better introduction of the evolutionary concepts/theories the whole story is built upon.
- Needs a short introduction about biology and life cycle of the pathogen since this has major implications for the interpretation of the genetic diversity and structure observed.
- Should contain clear hypothesis, research questions or goals of the study which can then be individually addressed in the discussion section.
- P3/9-10: "how "admixture variation" arises, and its genetic architecture within newly-adapted populations, is not well understood" – is this a "study aim" and does this get addressed in this manuscript?

Results:

- With 14 pages the result section is much too long. It contains tons of different abbreviations which are sometimes not consistent with figure legends. This makes the reading extremely difficult and influent. Authors should seek for possibilities to simplify this to a large extent. Specifically, the section about "swarm reconstruction" starting at page ten would profit from considerable shortening. It was also unclear why the subsections were grouped in seven points (e.g. you need to introduce these points and say what you are getting at.).

Discussion:

- Statements such as "Ours is a landmark achievement" always make me suspicious. It also contains a lot of hypothetical arguments and counterarguments from the authors with the same effect.
- For me the discussion should be placed in a broader context. What are now the true findings of the study and what's exactly the difference to previous studies with Darwin's finches or sticklebacks for example.

Materials and Methods:

- Compared to the results part, the M&M section is awfully short and lacks is not presented in sufficient detail: Misses a summary describing the data: where do the samples come from, when where they collected etc. – same with the reference genome: where did that come from?
- From the info given it is not possible to follow or reproduce the research approach and all the analyses performed. E.g. How many variants remained in the final SNP dataset? Were they bi-allelic? No mention of type of RIP-scan performed (P16.l1-2). The whole M&M section should be scanned by authors for missing relevant information and be complemented. In addition, a supplementary file with

3additional information could be compiled.

- Data availability: Custom scripts for the analyses should be made available to the public. E.g. ShinyHaplotypes.R

Presentation of figures:

The presentation of figures insufficient. Examples: Fig. 1: It is very difficult to connect the different clusters in 1c and the boxplots in 1d, i.e. the labels in 1d should also be used for the phylogenetic groups in 1c. The naming of phylogenetic groups is inconsistent. The number of strains usually is in parenthesis but not if there is only one strain (e.g. Wk2-1 (Lolium3)). The abbreviations used in 1d and in subsequent figures were never properly defined/introduced and in the text different abbreviations are used. Fig. 2: The colours in 2a lower panel are not appropriate. Different font for the % sign in the y-axis? Window position is not written consistently in 2a and 2c. 2c is not interpretable like this. There are too many different colours. The same is true for 2d even though it is a bit better. Fig. 4 is probably not necessary at all, contains typos such as "contibutions". Fig. 5. The box with node label and matching strains is too small. Fig. 6 the pie charts, font size and the two photos are too small.

Reviewer #2 (Remarks to the Author):

The submitted manuscript entitled "Recombination of standing variation in a multi-hybrid swarm drove adaptive radiation in a fungal pathogen and gave rise to two pandemic plant diseases" sought to explain how admixing of existing populations of the plant-pathogenic fungus, *Pyricularia oryzae*, resulted in two new pathogens, 'Wheat Blast' and 'Gray Leaf Spot'. The presented data support the idea that admixture of existing populations with very little subsequent divergence resulted in the emergence of these new pathogens. The manuscript offers an important example of how hybridization can have profound evolutionary consequences. As the authors note, recombination of standing variation is an evolutionary force that is sometimes eclipsed by an adaptationist focus on specific/new mutations. While the overarching drive of the study is supported and is of considerable significance to our understanding of evolution and pathogen emergence, some of the posited results are not well supported. The writing of the manuscript is also in need of extensive revision to be more clear, precise, and accessible to the broad target audience. I have detailed specific issues and my recommendations for how they should be approached below:

Overarching goal – pathogen emergence from population hybridization:

As noted above, the overarching goal of the study was to demonstrate that the emergence of two plant-pathogenic fungi was the result of admixture between existing populations. The authors use a large number of whole-genome sequences to demonstrate that the genomes of these newly emerging pathogenic populations are comprised of large contiguous chromosomal segments that are identical, or nearly identical, to potential donors in other populations. Within the pathogenic populations, the haplotypes of these donated segments show low divergence from source populations. These results together point to a recent hybrid swarm event and are consistent with the evidence used to support previous inference of such phenomena in other systems (e.g., Stukenbrock et al. 2012). There are, however, several points that could help strengthen this result.

4The manuscript currently relies on phylogenies of the BAC7 and MPG1 regions to explore haplotype divergences. However, the reason why these specific regions were selected is not immediately clear. I recommend that the authors include a depiction of the distribution of how many haplotypes, how diverged haplotypes, and how many segregating sites were found in each sliding window. Furthermore, it would be informative of later discussion of recombination load to provide information on how many SNPs were present in and between populations. At present, it is unclear if mention of “hundreds of thousands of alleles” reflects the number of SNPs. Population sample sizes and clonal fractions are also important metrics for informing patterns the authors observe (as high clonal fraction in some populations may skew ancestral probability estimates to extremes). Together these analyses will offer whole-genome context to the patterns evident in the above-mentioned phylogenies. While the authors offer comparison of haplotype divergences found in the new populations of interest to that found in the *Oryzae* population, a similar comparison looking at chromosomal ancestry would give important context for the results of this study. Finally, the authors should clarify why areas of no variation for chromosome 2 presented in Figure 2B do not appear to line up with areas of no variation depicted on the same chromosome in Figure 1A.

Main goal interpretation – inferring history of hybridization:

The authors present a logical series of steps to explain extant genomic structure of emerged pathogens. These steps are based upon haplotype analyses described above and rooted in sampling dates of isolates and are bolstered by dates generated from Bayesian inference of isolate evolution. The presented story does appear logical but is difficult to follow and does not adequately acknowledge the possibility of intermediate lineages that are now extinct or were unsampled. Problematically, most of the dendrograms/networks used for these inferences (e.g., Figure 5A and B) assume no recombination (note that reticulation in median spanning networks do not necessarily reflect recombination in the way that they do for the neighbor-net network). While it is not uncommon to use phylogenies that assume no recombination in recombining populations, the manuscript’s focus on recombination necessitates consideration of the limitations that these analyses impose. The authors presentation of the history leading to extant hybrids could be substantially bolstered by including an Ancestral Recombination Graph (which allows for recombination) and by utilizing programs that can infer recombination events (including from unobserved donors) directly from sequence data (e.g., ClonalFrameML). Indeed, inferences from ClonalFrameML or similar program could also bolster later discussion of avirulence genes (i.e., PWT6/PWT3). Together these analyses would substantially strengthen presented inferences and will also help to better depict the evolutionary history that is core to the story told by the manuscript.

Inferences of recombination and parasexuality

The authors infer recombination and re-assortment directly from neighbor-net networks and from chromosome-level haplotype analyses. Additionally, sex is evident from the presence of RIP mutations. The presented results are consistent with a history of recombination. However, the current approach is unlikely to detect recombination events that occur between very closely related lineages. This is particularly important for the authors suggestion that recombination happened during the formation of the extant populations, but not since. This inference is largely based on a lack of reticulation in Figure 5B. However, as noted above, reticulations in median spanning networks are not generally considered strong evidence of recombination. Detecting recombination from networks also often relies on sampling of both parents and offspring – something that is not a given in most large-

5scale sampling efforts. Inferences of recombination could again be strengthened using software that infers recombination events directly from sequence data (as noted above with ClonalFrameML). Additionally, analysis of mating-type gene frequencies would be useful to understand what matings would be possible in parental and offspring strains. Mating-type genes should be easily querriable from the authors existing sequence data and will help to confirm that the mating events they infer would be possible (as mating can only occur between fungi that have opposite mating-type genes). The authors also infer that one of the hybridization events occurred by a parasexual process. To support this, the authors rely on the physical size of recombination blocks from possible donors, suggesting that recombination blocks are larger than would be expected from sexual recombination and are instead more consistent with parasexuality. However, parasexuality in fungi is thought to be exceedingly rare in nature, with only a few well-described instances. Parasexuality is instead thought to largely be a laboratory phenomenon. Even in the species that do have observed parasexuality, there is not strong evidence of widespread gene conversions like those used here to invoke this process. I am skeptical of the authors approach for inferring parasexuality and am not aware of any other work that has inferred parasexuality with similar evidence -- I recommend removal of this inference from the manuscript.

Inferences of host range

The first section presented in the Results of this manuscript details the host-range of the isolates used in this study. However, it appears that no new pathogenicity experiments were conducted. Instead, inference of 'pathogenicity' seems to rely largely on the crop that isolates were originally sampled from. This metric of pathogenicity is not strong, as some isolates were found on crops that do not correspond to their population. Even with this crude metric, the majority of isolates in Figure S1 are not presented with host-pathogenicity data. Inferences of pathogenicity need better support and clearer indication of their origin. The current presentation of the data suggests there is some existing evidence that new populations have increased host range, but it is not completely clear how much of this is based on sampling vs actual ability of these organisms to infect different hosts.

Inferences of evolutionary processes in the formation of extant populations.

A major point of the manuscript is that pathogenic phenotypes in emerging populations are the result of recombination of standing variation and not the result of selection (see lines 13-16 of page 25). However, the authors do not actually test this hypothesis, but instead speculate that drift is the only answer to observed patterns. While inferences of selection from the site-frequency spectrum are likely precluded by the lack of diversity, haplotype frequencies could serve as the basis for comparisons of drift vs selection. The authors note that a single mating event can initiate hundreds of independent meioses. Based on the underlying biology of sex in this fungus it should thus be possible to roughly estimate the diversity generated in mating events the authors already infer. The probability of selection vs drift could then be computed using a binomial expression. While these estimates will be crude, they will be founded in a solid set of assumptions and thus inform the authors assertions of drift vs selection.

Overall writing and precision

The overall framing of the manuscript do not seem well suited to the large audience that is likely to find the results of this study interesting. In particular, the introduction does not prepare readers well to understand the largely-clonal nature of this species or the life history of the fungus described. The

6overall framing in an evolutionary context largely makes sense to me, but references to evolutionary theory are vague to the point of being largely inscrutable – particularly in the discussion. For example, the concluding sentence states “we believe that our findings point to an overlooked role for happenstance in creating situations that allow organisms to skirt rules that would normally hold evolution in check”. It is not immediately clear what ‘rules’ the authors are referring to and the statement seem at odds with Darwinian ideas that emphasize the role of genetic drift as an evolutionary force.

The manuscript also presents many ideas in non-scientific ways. The authors do not appropriately consider alternative hypotheses, often presenting results as fact. For example, the authors indicate that a large genomic insertion is from an unknown population (PoX). However, I see no evidence that this insertion is necessarily from a single population – PoX may reflect contribution from multiple unknown populations. Non-scientific language is also present throughout much of the manuscript. Phrases like ‘interrogating raw sequence reads’ do not reflect clear scientific analyses. Similarly, terms like ‘likelihood’ are used in ways that do not reflect their technical meaning. The phrase ‘fungal orgy’, while amusing, is not reflective of a scientific process. Overall, there is a need for extensive reworking of language throughout the manuscript.

The monikers used to describe haplotypes, isolates, and populations are often confusing. Haplotype names are very similar to population names while isolate designations sometimes are and sometimes aren’t. Isolates are sometimes referred to by their haplotypes, but sometimes by their DAPC-defined designation (these designations are again similar to haplotype names). One particularly problematic instance is on page 12 lines 5-13 where two strains are invoked by strain name and later compared to the PoL1 haplotype with no indication of the two strains haplotype. This issue is compounded by the fact that one of the cited strains (T47-3) does not occur in the isolate table. I recommend a reworking of how strains/haplotypes/populations are invoked.

In addition to the overall clarity of the manuscript, there are some issues with the methodologies presented. Missing in the methods section is a description of how isolates were collected – if, for example, multiple isolates were collected from a single sample, it could have important population-genetic implications because of non-independent sampling. Methodologies for transposon querying are completely missing. Concerningly, the authors use an in-house SNP calling software that has not been peer reviewed and the authors do not indicate if this software will be made available. Isolates used for various analyses and the metrics by which they were chosen are not immediately clear, and the authors refer to them using different nomenclature, sometimes even within the same figure caption (e.g., in Figure 2, I assume “test strains” are the same as “comparator strains”).

*****END*****

Author Rebuttal to Initial comments

RESPONSES TO REVIEWERS COMMENTS

7On behalf of my team, I thank the reviewers for their thoughtful and insightful comments. The revised manuscript is a greatly improved product thanks to their suggestions.

Original comments in black, responses in red.

REVIEWER 1:

The manuscript submitted by Rahnama et al. describes the evolution of two new pathogen lineages causing wheat blast and gray leaf spot of ryegrasses by using extensive genomic analyses. Authors convincingly show that the new lineages emerged through several admixture events between divergent host specific pathogen lineages and they state that admixed variation fuelled pathogen adaptive radiation. Unquestionable, the strength of the manuscript is the exceptional genomic dataset which was used to underline their hypothesis. The idea to use this dataset to underpin a very recent and widely discussed topic in evolutionary biology is striking. However, the manuscript revealed several major flaws that need to be addressed before publication in nature ecology and evolution. The manuscript is extraordinary difficult to read and understand. This is surely because of the very complex story of pathogen evolution authors want to sell and tell but to a large extent also due to the manuscript structure and the presentation of results. For example, the whole story is built on previous work about parallel evolution and admixture variation. These quite complex concepts and theories are introduced in no more than ten lines at the beginning of the introduction. Words and terms such as admixture variation, hybrid swarm, populations, adaptive radiation are not well introduced, defined and/or explained. (now explained in the introduction) The wording is sometimes inappropriate and contains a lot of personal statements and arguing such as “we believe...”, “we would simply point out...”. The presentation of figures is insufficient and the overall impression about the manuscript is “drafty”. The following major comments should help authors to improve the manuscript prior resubmission.

Introduction:

- needs a better introduction of the evolutionary concepts/theories the whole story is built upon.

Additional background has been provided

- Needs a short introduction about biology and life cycle of the pathogen since this has major implications for the interpretation of the genetic diversity and structure observed.

This information is included in the relevant sections

- Should contain clear hypothesis, research questions or goals of the study which can then be individually addressed in the discussion section.

The study had one main goal - to understand the rapid genetic diversification accompanying wheat blast/gray leaf spot evolution. Once the hybrid swarm was identified and its genetic architecture elucidated, the implications for evolutionary biology immediately became apparent.

- P3/9-10: “how “admixture variation” arises, and its genetic architecture within newly-adapted populations, is not well understood” – is this a “study aim” and does this get addressed in this manuscript?

8This was not a study aim - but findings provided new insights into these long-standing questions. This should now be clear after revisions.

Results:

- With 14 pages the result section is much too long.

Section has been shortened by eliminating the overly detailed description of avirulence gene dynamics. This information was included for the benefit of wheat blast researchers but detracts from the main thrust for a more general audience.

It contains tons of different abbreviations which are sometimes not consistent with figure legends. This makes the reading extremely difficult and inefficient. Authors should seek for possibilities to simplify this to a large extent.

We feel that the abbreviations are needed but have been clarified by improving consistency with figures.

Specifically, the section about “swarm reconstruction” starting at page ten would profit from considerable shortening.

Shortened by eliminating details on AVR gene dynamics.

It was also unclear why the subsections were grouped in seven points (e.g. you need to introduce these points and say what you are getting at.).

The subsections are now broken out as main sections instead.

Discussion:

- Statements such as “Ours is a landmark achievement” always make me suspicious.

This IS a landmark achievement - no other studies have even come close to the genetic resolution we have achieved in understanding an adaptive radiation.

It also contains a lot of hypothetical arguments and counterarguments from the authors with the same effect.

The arguments in question were eliminated because they were centered around when specific AVR gene alleles arose in relation to prior hypotheses for wheat blast evolution.

- For me the discussion should be placed in a broader context. What are now the true findings of the study and what's exactly the difference to previous studies with Darwin's finches or sticklebacks for example.

Hopefully, we have now captured the broader implications and novelty of our work. Even after the major advancements that have been made in other systems, four key questions remain. Our work provides new and key insights into all four.

Materials and Methods:

- Compared to the results part, the M&M section is awfully short and lacks is not presented in sufficient detail: Misses a summary describing the data

the reviewer must have missed the section describing data availability - all datasets were available for review) where do the samples come from, when where they collected etc. – same with the reference genome:

where did that come from?

All this information was present in Supplementary table 1.

- From the info given it is not possible to follow or reproduce the research approach and all the analyses performed.

Actually, all datasets and the scripts to run them were provided on the FigShare site.

E.g. How many variants remained in the final SNP dataset? Were they bi-allelic?

This information is now provided.

No mention of type of RIP-scan performed (P16.I1-2).

The whole M&M section should be scanned by authors for missing relevant information and be complemented. In addition, a supplementary file with additional information could be compiled.

The RIP analysis is the only part that was missing from the methods. It is now included.

- Data availability: Custom scripts for the analyses should be made available to the public. E.g. ShinyHaplotypes.R

These data and all scripts were available on FigShare at the time of the review.

Presentation of figures:

The presentation of figures insufficient. Examples: Fig. 1: It is very difficult to connect the different clusters in 1c and the boxplots in 1d, i.e. the labels in 1d should also be used for the phylogenetic groups in 1c.

This has now been done.

The naming of phylogenetic groups is inconsistent.

Corrected

The number of strains usually is in parenthesis but not if there is only one strain (e.g. Wk2-1 (Lolium3)).

This is intended but a note to that effect is included in the legend.

The abbreviations used in 1d and in subsequent figures were never properly defined/introduced and in the text different abbreviations are used.

The legend now describes the derivations of the lineage names.

Fig. 2: The colours in 2a lower panel are not appropriate.

These are the colors used for PoL and PoT throughout. It is not really essential to be able to see the individual traces. It's the wide variation in divergence that matters (deflection relative to the x-axis).

Different font for the % sign in the y-axis?

No, font is correct

Window position is not written consistently in 2a and 2c. 2c is not interpretable like this.

This has been corrected.

There are too many different colours.

Unavoidable - but the figure still gets across the point that the donor predicted by chromopainter, exhibits significantly lower haplotype divergences than the next "closest" candidate. The same is true for 2d even though it is a bit better.

Fig. 4 is probably not necessary at all, contains typos such as "contibutions".

11Corrected and moved to supplemental figures

Fig. 5. The box with node label and matching strains is too small.

Now enlarged

Fig. 6 the pie charts, font size and the two photos are too small.

Increased the sizes as recommended.

REVIEWER 2:

The submitted manuscript entitled "Recombination of standing variation in a multi-hybrid swarm drove adaptive radiation in a fungal pathogen and gave rise to two pandemic plant diseases" sought to explain how admixing of existing populations of the plant-pathogenic fungus, *Pyricularia oryzae*, resulted in two new pathogens, 'Wheat Blast' and 'Gray Leaf Spot'. The presented data support the idea that admixture of existing populations with very little subsequent divergence resulted in the emergence of these new pathogens. The manuscript offers an important example of how hybridization can have profound evolutionary consequences. As the authors note, recombination of standing variation is an evolutionary force that is sometimes eclipsed by an adaptationist focus on specific/new mutations. While the overarching drive of the study is supported and is of considerable significance to our understanding of evolution and pathogen emergence, some of the posited results are not well supported.

First, I'd like to preface my responses to reviewer 2's concerns by saying that I really appreciate the detailed and constructive comments. These are all valid comments for a "normal" population genetics/evolutionary biology study. However, the evolutionary scenario that underlies the emergences of wheat blast and gray leaf spot breaks several key rules and assumptions which means that most approaches to studying evolutionary phenomena are not appropriate/applicable. In fact, an extended finding from our study is that virtually none of the existing phylogenetic/population genetic tools are appropriate for the evolutionary scenario that gave rise to wheat blast/gray leaf spot. As a result, industry standard programs and statistical approaches for detecting population structure/admixture were not able to detect the admixtures we have documented here and gave wildly erroneous answers. Likewise, Ancestral Recombination Graph approaches and standard demographic modeling scenarios are not suitable.

We also fully recognize that some of our "conclusions" are not well-supported. However, the only reasons for this were limited sample availability and a lack of *de novo* nucleotide variation, which pose major challenges and actually make formal tests of several key conclusions impossible. Consequently, it is not possible to address this criticism

12because we are working with historical samples and used ALL of the ones that were informative on the questions at hand. We strongly feel that extrapolations most definitely should be made from our findings because, as the reviewer points out, "...the overarching drive of the study is supported and is of considerable significance to our understanding of evolution and pathogen emergence." Additionally, our findings provide a new and detailed contextual framework that will help to refocus and thereby advance thinking in evolutionary biology.

The writing of the manuscript is also in need of extensive revision to be more clear, precise, and accessible to the broad target audience. I have detailed specific issues and my recommendations for how they should be approached below:

The manuscript has been extensively edited for clarity.

Overarching goal – pathogen emergence from population hybridization:

As noted above, the overarching goal of the study was to demonstrate that the emergence of two plant-pathogenic fungi was the result of admixture between existing populations.

The goal was actually to understand how such high genetic divergence could have accumulated in just 60 years since wheat blast's emergence. Based on the resulting data, admixture between existing populations was the ONLY reasonable answer.

The authors use a large number of whole-genome sequences to demonstrate that the genomes of these newly emerging pathogenic populations are comprised of large contiguous chromosomal segments that are identical, or nearly identical, to potential donors in other populations. Within the pathogenic populations, the haplotypes of these donated segments show low divergence from source populations. These results together point to a recent hybrid swarm event and are consistent with the evidence used to support previous inference of such phenomena in other systems (e.g., Stukenbrock et al. 2012). There are, however, several points that could help strengthen this result.

We feel there is no need to "strengthen" our conclusions. There are no other reasonable explanations for the observed data.

The manuscript currently relies on phylogenies of the BAC7 and MPG1 regions to explore haplotype divergences.

However, the reason why these specific regions were selected is not immediately clear.

The launching point for the work described in the manuscript relied on phylogenies of eight genes - we selected these two to explore in more depth because they had the most discriminative power. This is now explained in the revised version.

I recommend that the authors include a depiction of the distribution of how many haplotypes, how diverged haplotypes, and how many segregating sites were found in each sliding window.

This information is explicitly plotted by ShinyHaplotypes. Number of different y-values at each window position represents the number of haplotypes and the number of segregating sites in each window (per 2000 variant sites) are

the y-axis values. In fact, the advantage of ShinyHaplotypes is that one is able to "see" these metrics and thereby easily understand the partitioning of variation, at any given position on the chromosome, in a single glance.

Furthermore, it would be informative of later discussion of recombination load to provide information on how many SNPs were present in and between populations.

The average SNPs divergence within populations was shown in Figure 1D. We have now included information on average between population divergence (~ 0.007).

At present, it is unclear if mention of "hundreds of thousands of alleles" reflects the number of SNPs.

This has been clarified by changing the text to "re-introduction of thousands of genes/alleles with divergent histories in at least 47 combinations." With a conservative estimate of 20% of genome segregating for an average of 2 alleles in 47 haplotypes, this represents $2,400 \text{ genes} \times 32 \text{ alleles} \times 47 \text{ combinations} = 225,600$ new combinations of gene alleles.

Population sample sizes and clonal fractions are also important metrics for informing patterns the authors observe (as high clonal fraction in some populations may skew ancestral probability estimates to extremes).

Together these analyses will offer whole-genome context to the patterns evident in the above-mentioned phylogenies.

Figure 1C and Figure 1D already provide the needed information. Population variation in the PoT/PoL lineages is clearly greater than in most populations and the pairwise nucleotide divergence values reveal the absence of clones. There were no clones in the ancestral populations. This has been clarified. The reviewer is correct that sample size affects probability estimates. This is why SHinyHaplotypes was developed to verify the ChromoPainter results.

While the authors offer comparison of haplotype divergences found in the new populations of interest to that found in the Oryzae population, a similar comparison looking at chromosomal ancestry would give important context for the results of this study.

While the suggested study would certainly be nice for understanding rice blast evolution, it is not useful for understanding the evolutionary history of PoT/PoL. We are already pushing the page limits with the present data.

Finally, the authors should clarify why areas of no variation for chromosome 2 presented in Figure 2B do not appear to line up with areas of no variation depicted on the same chromosome in Figure 1A.

Figure 1A shows a picture of wheat blast disease, so I cannot address this comment. If the reviewer means Figure 3A, then, haplotype PoT3 was not included as a comparator to focus the reader on chromosome regions showing almost no variation. This is now clarified in the legend. Also, note that by necessity, Shinyhaplotypes and Chromopainter cannot have equivalent x-axes (Mb versus window #), due to difficulties in rendering missing data in ShinyHaplotypes without destroying interpretability.

Main goal interpretation – inferring history of hybridization:

The authors present a logical series of steps to explain extant genomic structure of emerged pathogens. These steps are based upon haplotype analyses described above and rooted in sampling dates of isolates and are bolstered by dates generated from Bayesian inference of isolate evolution. The presented story does appear logical but is difficult to follow

The present version has been extensively edited for clarity

and does not adequately acknowledge the possibility of intermediate lineages that are now extinct or were unsampled.

The manuscript explicitly discussed not only the possibility of intermediate/extinct/unsampled lineages but based on the data, we could actually infer their existence - we just didn't use the term lineages, we referred to sibling progeny and counter-selected haplotypes.

Problematically, most of the dendrograms/networks used for these inferences (e.g., Figure 5A and B) assume no recombination (note that reticulation in median spanning networks do not necessarily reflect recombination in the way that they do for the neighbor-net network).

This is incorrect.. For the dendrogram in Figure 5A, we specifically selected chromosome regions that were inherited from the same ancestor (i.e. at T0, there were no SNPs). Therefore, recombination is only problematic if it occurs after the recombining chromosomes have acquired new SNPs in the regions under interrogation. The median spanning network shows that virtually all (if not all) SNPs arose after the swarm's recombinational activity had ceased, with all of the observed reticulations being due to rare secondary gene flow.

While it is not uncommon to use phylogenies that assume no recombination in recombining populations, the manuscript's focus on recombination necessitates consideration of the limitations that these analyses impose.

It's not just that it's not uncommon to use phylogenies that assume no recombination for recombining populations, it is in fact the rule! The vast majority of phylogenetic trees in the literature are based on assumption of no recombination when it is KNOWN that recombination does occurs and, in fact, the tree architecture tells us that recombination has occurred (thereby making the tree invalid).

The authors presentation of the history leading to extant hybrids could be substantially bolstered by including an Ancestral Recombination Graph (which allows for recombination) and by utilizing programs that can infer recombination events (including from unobserved donors) directly from sequence data (e.g., ClonalFrameML). (note

15ShinyHaplotypes infers recombination events from unobserved donors - that's how we identified the contribution from PoX).

The history of the extant hybrids needs no bolstering. There is no other explanation for the presented data without inferring novel evolutionary mechanisms and very badly disobeying Occum's razor in doing so.

Ancestral recombination graphs are just that: ANCESTRAL recombination graphs. They are not applicable in the present situation because the recombination events here are not ancestral - they are so recent that the number of SNPs available to inform the relevant programs are too small (169 SNPs total). Besides, based on the molecular dating and the haplotype network analyses, we can already predict that an ARG output would look like Figure 1 (if the relevant programs were even able to converge with such a small dataset). If one were to expand the SNP dataset, one would then be reconstructing the ancestral recombination events in the respective donor populations which is not relevant to the present study.

Indeed, inferences from ClonalFrameML or similar program (ChromoPainter IS similar to ClonalFrame in that it arrives at the same result using a different algorithmic implementation) could also bolster later discussion of avirulence genes (i.e., PWT6/PWT3). Together these analyses would substantially strengthen presented inferences and will also help to better depict the evolutionary history that is core to the story told by the manuscript. The evolutionary history that we have presented is practically complete - the only way it could be better depicted is if we had a member of the PoX lineage in hand.

I'm not sure that the reviewer appreciates the fact that, because these populations are so recently evolved, and we have the two earliest founder strains in hand, we basically have forensic-level understanding of the origin of every SNP and, by extension, every AVR gene allele in every genome.

Inferences of recombination and parasexuality

Here, I'm starting to suspect that maybe we confused the reviewer because the study talks about two recombinational episodes: i) the demonstrated event(s) that occurred in a swarm setting; and then ii) possible events occurring in the field after host infection and, therefore, possibly influenced by selection. All available evidence points to recombination having occurred **ONLY** in the swarm, and not in "the field."

The authors infer recombination and re-assortment directly from neighbor-net networks and from chromosome-level haplotype analyses. Additionally, sex is evident from the presence of RIP mutations. The presented results are consistent with a history of recombination. However, the current approach is unlikely to detect recombination events that occur between very closely related lineages.

This is incorrect. Because so few SNPs have arisen since the foundation of the new population(s), a single SNP is diagnostic, due to the extreme unlikelihood that the same SNP arose independently when there are only 5 or so SNPs total across a 2 Mb region of the genome. For this reason, we essentially have the ability to detect recombination based on only two SNPs. The median-spanning network does detect the putative recombination events and represents them as two or more paths between nodes.

Recombination between closely related HAPLOTYPES is easily detected because, by definition, they contain at least one chromosome segment (usually more) that was donated by a different parent.

This is particularly important for the authors suggestion that recombination happened during the formation of the extant populations, but not since. This inference is largely based on a lack of reticulation in Figure 5B. However, as noted above, reticulations in median spanning networks are not generally considered strong evidence of recombination. Detecting recombination from networks also often relies on sampling of both parents and offspring – something that is not a given in most large-scale sampling efforts.

The recombination network traces paths through every SNP that occurs between isolates and accounts for EVERY SNP in the dataset. Reticulations show up when paths between three isolates are incongruent with the established paths for each pair, and occur when: two isolates recombine after divergence; if an isolate experiences external gene flow; or if back-mutation occurs. All reticulations were checked (by interrogating individual SNPs) and were shown to be due to external gene flow.

Inferences of recombination could again be strengthened using software that infers recombination events directly from sequence data (as noted above with ClonalFrameML).

Again, not possible due to small size of dataset - ClonalFrameML builds maximum likelihood trees across different regions of the genome, and these trees are going to be meaningless when most isolates vary by ~10 SNPs.

Additionally, analysis of mating-type gene frequencies would be useful to understand what matings would be possible in parental and offspring strains. Mating-type genes should be easily queryable from the authors existing sequence data and will help to confirm that the mating events they infer would be possible (as mating can only occur between fungi that have opposite mating-type genes).

Mating types and frequency data were already presented in the manuscript (Supplementary table 2, and discussion) and were discussed as a possible reason for the observed lack of mating among epidemic isolates). Where known, mating-type information has now been added to Figure 5.

The authors also infer that one of the hybridization events occurred by a parasexual process. To support this, the authors rely on the physical size of recombination blocks from possible donors, suggesting that recombination blocks are larger than would be expected from sexual recombination and are instead more consistent with parasexuality. However, parasexuality in fungi is thought to be exceedingly rare in nature, with only a few well-described instances.

This is one reason NOT to remove the data because it provides evidence that parasex might occur more commonly than is currently suspected.

Parasexuality is instead thought to largely be a laboratory phenomenon.

This implies that fungi evolved hyphal fusion and mitotic exchange capability, as well as vegetative incompatibility, in anticipation of the event that someone might capture them and try to force their interaction in the lab!

Even in the species that do have observed parasexuality, there is not strong evidence of widespread gene conversions like those used here to invoke this process.

Please cite a study where this has been investigated. I am not aware of any that has even looked for gene conversions (in population data). This is one of several key advances in the present work.

I am skeptical of the authors approach for inferring parasexuality and am not aware of any other work that has inferred parasexuality with similar evidence.

It is widely accepted that meiotic conversion tracts are short, and mitotic conversions are much longer.

-- I recommend removal of this inference from the manuscript.

Based on the overall (low) crossover frequency, the large number of introgression blocks ranging from 10 kb to 50 kb can only be reasonably explained by mitotic gene conversions - for which there is abundant evidence for long tract lengths. Otherwise, we have to start inferring novel biology. Again, we opt for Occum's razor.

Inferences of host range

The first section presented in the Results of this manuscript details the host-range of the isolates used in this study. However, it appears that no new pathogenicity experiments were conducted. Instead, inference of 'pathogenicity' seems to rely largely on the crop that isolates were originally sampled from. This metric of pathogenicity is not strong, as some isolates were found on crops that do not correspond to their population. Even with this crude metric, the majority of isolates in Figure S1 are not presented with host-pathogenicity data. Inferences of pathogenicity need better support and clearer indication of their origin. The current presentation of the data suggests there is some existing evidence that new populations have increased host range, but it is not completely clear how much of this is based on sampling vs actual ability of these organisms to infect different hosts.

The repeated sampling of wheat blast and gray leaf spot pathogens from Avena, Bromus, Eleusine, Hordeum Melinis, Urochloa, etc. over multiple years and in geographically dispersed regions attests to the expanded infection capability of these new host-adapted forms. No further testing is necessary. Besides, "pathogenicity" experiments are meaningless for this fungus. There has been a long history of documenting cross-infectivity using inoculation assays. However, with the exception of fungal isolates of wheat blast and gray leaf spot, instances of cross infection in nature are extremely rare. In other words, fungal isolates collected from a given host almost invariably group phylogenetically with isolates from the same host (see Figure 1C).

Inferences of evolutionary processes in the formation of extant populations.

A major point of the manuscript is that pathogenic phenotypes in emerging populations are the result of recombination of standing variation and not the result of selection (see lines 13-16 of page 25). However, the authors do not actually test this hypothesis, but instead speculate that drift is the only answer to observed patterns. While inferences of selection from the site-frequency spectrum are likely precluded by the lack of diversity, haplotype frequencies could serve as the basis for comparisons of drift vs selection.

We use logical arguments to conclude that host selection was not operational between the serial admixture events: i) Fig. 5B showed that the only variants not acquired from standing variation most SNPs accumulated independently of haplotype groups. Again, if we start trying to infer selection as an This cannot be tested statistically because so few isolates were collected in the early years of the outbreak. Site frequency spectrum analysis is not applicable.

The authors note that a single mating event can initiate hundreds of independent meioses. Based on the underlying biology of sex in this fungus it should thus be possible to roughly estimate the diversity generated in mating events the authors already infer. The probability of selection vs drift could then be computed using a binomial expression. While these estimates will be crude, they will be founded in a solid set of assumptions and thus inform the authors' assertions of drift vs selection.

Here it is critical to understand that we are not saying that host selection hasn't occurred - it clearly has. What we are saying is that it does not appear to have been at play during the formation of the haplotypes that are pathogenic to wheat, *Lolium* and other hosts. In other words, the absence of PoSt introgressions in the PoL1 lineage is not due to

19gradual and recurrent loss of those sequences through intervening cycles of infection and mating. This is because we see no evidence for recombination in the epidemic population (i.e. once new SNPs had arisen), which implies that all sexual activity ceased once isolates had exited the swarm, presumably because the latter provided a genetic and physiological environment that was uniquely conducive to mating.

Clearly, if we were to use statistics to measure selection versus drift, the data would support selection. The differences are obvious based on the visuals alone. HOWEVER, we are arguing that this selection occurred only after all matings involved in the creation of the "favored" haplotypes were complete. In other words, the swarm activity generated thousands (millions?) of untested recombinants, and a small number happened to become adapted to a new host(s) entirely through quasi-random mating.

Overall writing and precision

The overall framing of the manuscript do not seem well suited to the large audience that is likely to find the results of this study interesting. In particular, the introduction does not prepare readers well to understand the largely-clonal nature of this species or the life history of the fungus described. The overall framing in an evolutionary context largely makes sense to me, but references to evolutionary theory are vague to the point of being largely inscrutable – particularly in the discussion. For example, the concluding sentence states “we believe that our findings point to an overlooked role for happenstance in creating situations that allow organisms to skirt rules that would normally hold evolution in check”. It is not immediately clear what ‘rules’ the authors are referring to and the statement seem at odds with Darwinian ideas that emphasize the role of genetic drift as an evolutionary force.

The manuscript also presents many ideas in non-scientific ways. The authors do not appropriately consider alternative hypotheses, often presenting results as fact. For example, the authors indicate that a large genomic insertion is from an unknown population (PoX). However, I see no evidence that this insertion is necessarily from a single population – PoX may reflect contribution from multiple unknown populations.

All PoX segments are phylogenetically most closely related to the PoO/PoS/PoLe lineages. This is most consistent with PoX sequences having come from a SINGLE population related to these lineages. The likelihood that multiple "PoX" donor populations equally divergent from PoO/PoS/PoLe contributed to PoT/PoL1 evolution is negligible and, therefore, disobeys Occum's razor. This is now explained in the text.

Non-scientific language is also present throughout much of the manuscript. Phrases like ‘interrogating raw sequence reads’ do not reflect clear scientific analyses.

Interrogate: " obtain data from (a computer file, database, storage device, or terminal)."

Similarly, terms like ‘likelihood’ are used in ways that do not reflect their technical meaning.

Likelihood: the state or fact of something's being likely; probability.

The phrase ‘fungal orgy’, while amusing, is not reflective of a scientific process. Overall, there is a need for extensive reworking of language throughout the manuscript.

This phrase has been removed.

The monikers used to describe haplotypes, isolates, and populations are often confusing. Haplotype names are very similar to population names while isolate designations sometimes are and sometimes aren't.

Manuscript has been edited for consistency. Naming convention is described in the legend to Figure 1.

Isolates are sometimes referred to by their haplotypes, but sometimes by their DAPC-defined designation (these designations are again similar to haplotype names). One particularly problematic instance is on page 12 lines 5-13 where two strains are invoked by strain name and later compared to the PoL1 haplotype with no indication of the two strains haplotype. This issue is compounded by the fact that one of the cited strains (T47-3) does not occur in the isolate table. I recommend a reworking of how strains/haplotypes/populations are invoked.

These omissions have been corrected.

In addition to the overall clarity of the manuscript, there are some issues with the methodologies presented. Missing in the methods section is a description of how isolates were collected – if, for example, multiple isolates were collected from a single sample, it could have important population-genetic implications because of non-independent sampling.

All samples are independent and this now mentioned in the methods.

Methodologies for transposon querying are completely missing.

These results are no longer reported/discussed and have been removed.

Concerningly, the authors use an in-house SNP calling software that has not been peer reviewed and the authors do not indicate if this software will be made available.

We used an in-house SNP caller because we have found that the industry-standard programs (BWT/GATK) have inherent flaws that produce unacceptably inaccurate SNP calls in ALL fungal datasets that we have analyzed. Our caller was developed specifically to avoid these problems. SNP calls based on BWT/GATK routinely produce up to 40% false calls; while at the same time completely missing up to 25% true calls. This is because chromosome regions that are highly divergent between the reference genome and the query either fail to align completely, or align at read depths unrelated to their true depth. These problems are due to: i) repeat sequences that have diverged due to RIP. SNPs in repeats should NOT be included in any datasets for evolutionary or population studies; or ii) sequences that are highly diverged due to introgression. These SNPs absolutely SHOULD be included in evolutionary/population studies, considering that variation is the primary building block for evolution/population diversification. We have fully validate our pipeline in two ways - calling SNPs between independent assemblies of the same genome from non-overlapping, sub-samplings of reads from the same sequence run (~20 false calls / megabase); and by validating each variant site by interrogating raw reads (XX% SNPs supported). Software availability was mentioned in the relevant forms with the original submission but is now explicitly stated in the manuscript.

21Isolates used for various analyses and the metrics by which they were chosen are not immediately clear, and the authors refer to them using different nomenclature, sometimes even within the same figure caption (e.g., in Figure 2, I assume “test strains” are the same as “comparator strains”).

The test strain is the strain being tested and the comparator is the strain against which it is being compared. Test strains were selected as exemplars of general trends, while all strains from each indicated lineage were included as the comparators. The test strain is only a comparator when it is not the "test" and its lineage has been selected for comparison. This is now clarified in methods.

Decision Letter, first revision:

20th April 2022

Dear Professor Farman,

Your revised Article, "Recombination of standing variation in a multi-hybrid swarm drove adaptive radiation in a fungal pathogen and gave rise to two pandemic plant diseases" has now been seen by Reviewer #2. Unfortunately, Reviewer #1 was not available to look at this revision.

You will see from their comments copied below that Reviewer #2 believes their comments have not been addressed to satisfaction. I have read your responses to both reviewers as well as the revised manuscript and agree that their comments have not been addressed to satisfaction. The revision lacks the necessary toning down and discussion of caveats as well discussion of alternative interpretations of the results. Both reviewers initially thought that the data were interesting, so we have decided to give you one more chance to address the multiple comments to satisfaction. We urge you to consider again the alternative analyses suggested by Reviewer #2. In addition, you must discuss caveats associated with the multiple analyses and conclusions in your study, as explained by Reviewer #2 in their two reports. Also, we agree with Reviewer #1 that the Methods section should be expanded to include details of methods that allow to reproduce the analyses--referring to appended datasets is not sufficient. Please note that there is no limit to the length of the Methods section.

If you wish to submit a substantially revised manuscript following this guidance, please bear in mind that we will be checking the revision and new response to reviewers letter carefully, and won't approach reviewers again unless all points have been addressed to satisfaction. We should also recruit a new reviewer.

If you choose to revise your manuscript taking into account all reviewer and editor comments, please

22highlight all changes in the manuscript text file in Microsoft Word format.

* Include a "Response to reviewers" document detailing, point-by-point, how you addressed each referee comment. If no action was taken to address a point, you must provide a compelling argument. This response will be sent back to the referees along with the revised manuscript.

* If you have not done so already we suggest that you begin to revise your manuscript so that it conforms to our Article format instructions at <http://www.nature.com/natecolevol/info/final-submission>. Refer also to any guidelines provided in this letter.

[REDACTED]

If you wish to submit a suitably revised manuscript we would hope to receive it within 6 months. If you cannot send it within this time, please let us know. We will be happy to consider your revision so long as nothing similar has been accepted for publication at Nature Ecology & Evolution or published elsewhere.

Nature Ecology & Evolution is committed to improving transparency in authorship. As part of our efforts in this direction, we are now requesting that all authors identified as 'corresponding author' on published papers create and link their Open Researcher and Contributor Identifier (ORCID) with their account on the Manuscript Tracking System (MTS), prior to acceptance. This applies to primary research papers only. ORCID helps the scientific community achieve unambiguous attribution of all scholarly contributions. You can create and link your ORCID from the home page of the MTS by clicking on 'Modify my Springer Nature account'. For more information please visit www.springernature.com/orcid.

Thank you for the opportunity to review your work.

[REDACTED]

Reviewers' comments:

Reviewer #2 (Remarks to the Author):

The resubmission of the manuscript entitled 'Recombination of standing variation in a multi-hybrid swarm drove adaptive radiation in a fungal pathogen and gave rise to two pandemic plant diseases' has not addressed the concerns I raised in the original review about the strength of evidence supporting claims and the writing. While the writing has been made somewhat clearer, there are still pervasive issues in the grammar, flow, and content. Perhaps because of what improvements have been made to the writing, the revised manuscript appears to have even more glaringly-unsupported speculative claims that are presented as fact than the original submission. The evidence supporting claims of the manuscript are not robust and the authors proclivity for unbridled extrapolation to other organisms and to scientific theory itself are inappropriate. I detail the overarching nature of these issues below.

In their response to review the authors say "We also fully recognize that some of our 'conclusions' are not well-supported". Despite the fact that they agree with the suggestion from my original review that there are issues of support for many of their claims, the authors have dismissed all of my suggestions to further support their claims. The authors justify this dismissal by saying that other methods for offering support to the claims are not 'valid' for the system as the underlying biology in this system would 'break key rules and assumptions'.

The authors are correct that largely asexual species like the one that is the focus of this study do violate assumptions of many population genomic tests. However, this issue is not unique to this study: many fungi have clonal or semi-clonal population structures that violate these assumptions. Because of this, the most robust studies will perform many analyses and interpret the results in the context of violated assumptions (many tests have predictable results that would result from certain underlying population demographics).

Although the authors reject further analyses to support their data citing violated assumptions, they have performed some analyses where assumptions are violated. Despite their proclaimed commitment to not violating assumptions, the authors have not included any nuanced interpretation that acknowledges the violations that do exist in their study. It is difficult to reconcile the refusal to perform additional analyses because of potentially violated assumptions with their rejection of my suggestion to discuss already-violated assumptions. For example, the phylogenetic analyses and network analyses performed by the authors assume no recombination. Given the focus of the study on recombination, I suggested the addition of some discussion of this violated assumption. The author's respond by indicating that it was actually the 'rule' to violate this assumption and have not added necessary discussion. The idea that violation of this assumption is the 'rule', is simply false. Many studies that focus on recombination acknowledge the fact that phylogenies like those used by the authors assume no recombination, and cautiously interpret results in that context. Furthermore, alternative approaches to networking like a neighbor-net network that allow for recombination are widely used by other studies.

In my original review I pointed out that the authors inferences of parasexuality are not robust. The

24crux of their approach is extrapolating from other studies' inferences of the length of meiotic vs mitotic conversions observed in model organisms in vitro. Based on the identification of shorter-length conversions, they infer mitotic mechanisms are more likely and thus invoke parasexuality. This approach is 1) unprecedented and unvetted 2) There is no evidence of this conversion pattern in natural systems where parasexuality is known to occur (e.g., in the best studied natural examples of parasexuality from *Cryphonectria parasitica* there are no reports of this pattern) 3) based on extrapolation of results from entirely different organisms in a controlled laboratory environment. Despite these glaring limitations, the authors present inferences of parasexuality as fact and do not offer any suggestion of doubt surrounding these inferences.

Instead of thoughtful discussion of possible alternative hypotheses that readers may consider to explain the results of the study, the authors spend nearly entire discussion lauding their results and wildly extrapolating to other systems. While I pointed out several possible instances where careful discussion of alternative hypotheses would be appropriate, the authors respond by invoking 'Occom's razor' rather than perform additional analyses or adding any discussion. Occom's razor is a useful heuristic to choose a hypotheses, it is not a robust way to falsify a hypothesis or to dismiss alternative hypotheses. For example, the authors do not ever consider the potential that selection may be an important factor in their results, but that selection manifests in recombination load. As a result, the vast majority of recombination events would be selected against, and the inferred events may be rare examples of combinations that are advantageous. This alternative hypothesis marks a fundamentally-different explanation to what the authors posit as fact.

Finally, while there does seem to have been some clarification of the language and flow in this document, problems with writing persist. Speculation presented as fact, unscientific language, and grammatical errors are widespread in the MS. Several of these issues were noted in my original review. I am hesitant to spend the time pointing out specifics of these as I highlighted several in my original review that the authors have not addressed. For example, I previously noted that the statement "Our findings point to an overlooked role for happenstance in creating situations that allow organisms to skirt rules that would normally hold evolution in check" is unclear, does not clearly reflect what was found, and incorrectly misrepresents evolutionary theory (I previously pointed out that even Darwin emphasized the importance of genetic drift). This statement has not been fixed, contextualized, or clarified and now appears to occur twice on lines 42, 586.

Author Rebuttal, first revision:

Response to review #2

Here we show each reviewer criticism in black and the corresponding response in red.

The resubmission of the manuscript entitled ‘Recombination of standing variation in a multi-hybrid swarm drove adaptive radiation in a fungal pathogen and gave rise to two pandemic plant diseases’ has not addressed the concerns I raised in the original review about the strength of evidence supporting claims and the writing. While the writing has been made somewhat clearer, there are still pervasive issues in the grammar, flow, and content. Perhaps because of what improvements have been made to the writing, the revised manuscript appears to have even more glaringly-unsupported speculative claims that are presented as fact than the original submission. The evidence supporting claims of the manuscript are not robust and the authors proclivity for unbridled extrapolation to other organisms and to scientific theory itself are inappropriate. I detail the overarching nature of these issues below.

Owing to the need to shorten the manuscript, the present revision focuses specifically on reporting evidence supporting the conclusion that wheat blast and gray leaf spot co-evolved in a multi-hybrid swarm, and that their evolution was part of a broader adaptive radiation. We have avoided overly extrapolative discussion and, so, most of the criticisms about "unbridled extrapolation to other organisms and scientific theory" should have been addressed.

In their response to review the authors say “We also fully recognize that some of our ‘conclusions’ are not well-supported”.

This quotation is taken out of context, we agreed that we make a small number of inferences (note: we did not use the term "conclude," or imply it) about specific details that are not well supported, but this is only because there is no way to apply formal tests to one time, historical occurrences. Instead throughout the manuscript, we drew conclusions when we had overwhelming supporting evidence. Everywhere else, we used terms such as "infer, speculate, suggest, etc." which should have made it clear that these inferences had less support.

Despite the fact that they agree with the suggestion from my original review that there are issues of support for many of their claims (not strictly true, see response above), the authors have dismissed all of my suggestions to further support their claims. The authors justify this dismissal by saying that other methods for offering support to the claims are not ‘valid’ for the system as the underlying biology in this system would ‘break key rules and assumptions’.

The specific "claims" to which the reviewer is referring need no further support (see below). Indeed, in their first review, the reviewer stated "While the overarching drive of the study is supported and is of considerable significance to our understanding of evolution and pathogen emergence, some of the posited results are not well supported." The additional experiments that were suggested to further support our claims are ones that they agreed were already supported, and for which we already have 12 specific lines of evidence that they are correct (see below). The fact of the matter is that there is only one viable explanation for the data presented - that wheat blast and gray leaf spot co-evolved about 50 years ago in a multi-hybrid swarm. If the reviewer can come up with any alternative explanations (that take into account all the data we presented), we will be happy to perform additional tests to rule out those alternatives.

In our original response, we pointed out that the experiments suggested by the reviewer - while perfectly valid for a "normal" system - are inapplicable in the present situation. Here, we would like to emphasize that the confounding factors are manifold and include the multi-hybrid nature of the population, the retention of multiple alleles at most loci, the existence of a huge founder effect, and the accumulation of VERY few mutations since the main hybridization event. These factors cause most population genomic analyses to arrive at erroneous solutions because the relevant tools explicitly assume tree-like evolution with occasional cross-branch gene flow.

The specific reasons we have not followed these suggestions are as follows:

Suggestion 1) perform chromopainting for rice blast isolates. This analysis of the relatively old rice blast population would not provide further support for our claims -whatever the outcome of such an experiment, the findings would have ZERO influence on the conclusions of the present study. The word and figure limits also preclude inclusion of such data.

Suggestion 2) The reviewer's second suggestion reads: "The authors' presentation of the history leading to extant hybrids could be substantially bolstered by including an Ancestral Recombination Graph (which allows for recombination) and by utilizing programs that can infer recombination events (including from unobserved donors) directly from sequence data (e.g., ClonalFrameML)."

First, we rejected this suggestion because we felt that our inference of recombination needs no further bolstering. The original submission already provides multiple lines of evidence that WB and GLS co-evolved through recombination (Figures 2A, 2C, 2D, 3A, 3B, 4; Table S3; Supplemental Figures S2, S4, S7). In fact the data we presented are far more substantial (and more informative about the hybrid population) than were presented in the PNAS citation that the reviewer cited as an exemplar.

We also previously explained that we rejected the suggestions to use ARGweaver and ClonalFrameML because they are not appropriate for the following reasons:

a) ARGweaver

ARGweaver is not useful to us because the algorithmic implementation of this tool makes it non-applicable to the wheat blast population. Specifically, the recombination events driving wheat blast evolution are not ancestral they are all contemporary. Therefore, the evolutionary histories of WB/GLS undermine a key assumption in the ARGweaver algorithm which requires that the mutation rate be much greater than the recombination rate, and this will cause ARGweaver to arrive at erroneous results. Operationally speaking, ARGweaver identifies crossover points and then infers recombination dates based on divergence between non-recombinant chromosome segments between the crossovers. However, because wheat blast and gray leaf spot are so recently evolved most non-recombinant windows will lack SNPs resulting in inferred recombination dates of 0 years.

This is not just theory: to demonstrate these points, we ran ARGweaver and it produced a graph looking just like the predicted one, but the estimated recombination dates were too low (0 - 8 years). The key point here is that the resulting ARG supported our findings but we knew they would before even running the program because we have a COMPLETE understanding of how the population (and hence, the input dataset) is structured. The ARGweaver data are not included in the revised manuscript because they don't provide any additional insights and because of figure/word limits but, for the reviewer's benefit, they are accessible at the URL:

<https://github.com/drdna/WheatBlastEvolution/tree/main/ARG>

b) ClonalFrameML. The statistical model underlying ClonalFrame is inappropriate for WB/GLS evolution because it is designed to identify clonal segments of the genome. In doing, so it makes a fundamental assumption that is immediately violated by the WB population because it did not arise via clonal propagation. From the original publication:

28“When δR ..(the product of recombination rate and average introgression length) .. is greater than one, there is a significant chance that recombination happened more than once at any genomic position for the longer branches of the phylogeny, but this is not accounted for in the ClonalFrame model which considers that each position is either imported or not.” Underline added for emphasis.

After a brief discussion of implications, the authors of the ClonalFrameML publication concede that:

“..(in) promiscuous species the signal of clonal inheritance is rapidly lost so that models of pure admixture may be more appropriate, such as the Structure and FineStructure models where linkage disequilibrium is caused only by linkage along the genome.”

That ClonalFrameML is inappropriate for the WB system is based not only in theoretical arguments but on re-analysis of ClonalFrame output in a publicly-available WB dataset (<https://github.com/Burbano-Lab/wheat-clonal-lineage>) that is linked from a recent preprint on BioRxiv (<https://www.biorxiv.org/content/10.1101/2022.06.06.494979v1>). Careful examination of those data reveals that ClonalFrameML failed to identify any admixture segments. This is because the distribution and composition of these segments (multiple "foreign" alleles at many loci) is not baked into the ClonalFrameML statistical model.

The authors are correct that largely asexual species like the one that is the focus of this study do violate assumptions of many population genomic tests. However, this issue is not unique to this study: many fungi have clonal or semi-clonal population structures that violate these assumptions.

In the original response to reviews, we explained the violated assumptions and the clonal or semi-clonal nature of the fungus is NOT one of them. NONE of the experiments we performed violate any assumptions regarding clonality or sexuality.

Because of this, the most robust studies will perform many analyses and interpret the results in the context of violated assumptions (many tests have predictable results that would result from certain underlying population demographics).

Our study did include MANY analyses and the reason why violated assumptions were not discussed is because there weren't any - all data from all analyses were perfectly consistent. This is the main reason we were confident in saying that our main "conclusions" need no further support.

Here is a summary of the analyses reported this study (and a few unreported analyses), and how each supported the main conclusions:

i) Published phylogenetic data (Gladioux et al. 2018) and Fig.S2 identify multiple alleles at several genomic loci.

ii) Published and present data show that each allele was identical in sequence to one found in another host-specialized population of *P. oryzae*.

iii) A phylogenetic analysis performed in the present study expanded on those observations (Fig.S2) and showed evidence of recombination between linked loci.

iv) Chromopaintings predicted the introgression of alleles from five different populations. The inferred donor populations precisely match those previously identified in the phylogenetic analyses (Fig. 4).

v) the same donor populations were inferred by:

ShinyHaplotypes (present study) (Fig. 2)

Phylogenetic trees based on mitochondrial DNA (not included due to word/figure limits)

Nearest Neighbor analysis (not included due to word/figure limits)

STRUCTURE (not included due to word/figure limits)

ARGweaver (not included due to word/figure limits)

Topological Weighting by Iterative Sampling of SubTrees (TWISST) (data not included due to word/figure limits).

vi) The chromopaintings/ShinyHaplotypes results perfectly predicted the ancestries for:

phylogenetic marker loci used by Gladieux et al. (2018)

all PWT6 and PWT3 alleles

mating-type alleles

the MoT3 and C17 diagnostic markers

dozens of other random markers

vii) ShinyHaplotypes confirmed the chromosome ancestries inferred by ChromoPainter (Fig. 2)

viii) ShinyHaplotypes revealed that the haplotypes (= sequences) of chromosome segments in the PoT/PoL1 isolates were nearly identical to those in at least one member of the predicted donor lineage (Fig. 2).

ix) ShinyHaplotypes revealed that chromosome segments predicted to be inherited from the same donor isolate had identical or nearly identical sequences among all isolates that inherited those segments (Fig. 3).

x) The multi-hybrid swarm model predicted the occurrence of sequential admixture events, with the expectation that the "early" PoL1 and PoT populations might contain strains representing intermediate admixtures. When we analyzed a number of isolates collected in and soon after the first disease outbreaks, we were successful in identifying the predicted intermediates (PoL1-1 and PoT1).

xi) Chromopainting of the intermediate isolates (PoL1-1 and PoT1) revealed chromosome architectures that definitively identify them as original founders (Fig.4) because:

- All of the crossovers between chromosome segments from different admixture donors can be found in PoL1/PoT descendants, with some crossovers being present in all, or nearly all, population members.
- The (PoE, PoU1 and PoX) donor contributions found in the founder strains account for most (if not all) of the respective donor contributions found in the entire PoL1/PoT population.

xii) Detailed crossover analyses identified the reciprocal products of individual meiotic exchanges, thereby confirming the participation of sibling progeny in matings - a defining feature of hybrid swarms.

Although the authors reject further analyses to support their data citing violated assumptions, they have performed some analyses where assumptions are violated. Despite their proclaimed commitment to not violating assumptions, the authors have not included any nuanced interpretation that acknowledges the violations that do exist in their study. It is difficult to reconcile the refusal to perform additional analyses because of potentially violated assumptions with their rejection of my suggestion to discuss already-violated assumptions. For example, the phylogenetic analyses and network analyses performed by the authors assume no recombination.

While we do accept partial blame for the initial criticism related to possible recombinational influences because the details of how the study was constructed to minimize such issues was only explained in the methods section, in our response to the first review, we explained very clearly that the way we constructed the dataset made us fully justified in assuming no recombination. We also pointed out that virtually all of the SNPs used for dating arose after recombinational activity had ceased, and that secondary introgressions capable of influencing the estimates were rare. The revised manuscript now explains the principle of the phylogenetic analysis and construction of the dataset in the text body. However, it is critical that the reviewer understand how we essentially "immunized" our phylogenetic analyses to recombinational influences, so we explain in greater detail here:

By their very nature, distance trees, as shown in Figure 1a, make no implicit assumptions about recombination (although it is possible a viewer might erroneously try to do so). To minimize this

possibility, we interpreted the tree specifically in terms of the genetic divergence between and within host-specialized populations. Notably, the mechanism by which that divergence arises has no bearing on the conclusion we draw from that analysis, namely that divergence is greater in the PoT/PoL1 populations - this is the very observation that told us there was more to WB/GLS evolution than the simple losses/mutation of host-specificity genes.

With regard to the phylogenetic study used for molecular dating, the main aim of the study was to estimate the date when the original PoL1-1 founder evolved. This would be the date when the isolate arose in the PoE1 x PoU1 mating. To this end, we used the chromopaintings in Figure 4 as a guide and were extremely careful to include only those SNPs that were in chromosome regions that all WB/GLS strains appeared to have inherited from the founder individual. By doing this, we eliminated influences from recombination events that occurred both prior to, and within, the founder cross. This is because the dating is now based on a clock that is reset at the admixture date (i.e. there are 0 mutations at T0). Note: For those not used to thinking about how individual SNPs and recombination events potentially affect dating estimates, these points may take some time, and careful thinking, to conceptualize.

By "resetting the clock," the only way mutations could influence our dating estimates is if the dataset contains SNPs that were not inherited from the founder but immigrated on a chromosome segment acquired via a secondary introgression. In the original response to the reviewer, we pointed out that these were rare. In fact, only 3 SNPs out of the 169 used for dating fell into this category. It did not seem worth discussing such a small error rate, when it would alter our estimate by <1 year.

Nevertheless, we took the reviewer's concerns very seriously and decided to add another chromosome segment to the dataset to convince them that we are not overlooking recombination events. This not only increased the probability of detecting chromosome reassortment from 0.5 to 0.75 per pairwise strain comparison but, by enriching the dataset from 169 to 422 SNPs, we improved the resolution of our dating estimates so that we were not only able to date the founder event (within the same range as we previously reported) but we arrived at a reasonable date for the swarm's formation too. Equally importantly, and in regard to the reviewer's original concerns, we identified only 15 recombinant SNPs within the 422 SNP dataset which resulted in the over/underestimation of split dates for only 8 out of 31 haplotypes, and did not affect the main conclusions. Nevertheless, these recombination events and their influences on the analysis are discussed in the revision.

Given the focus of the study on recombination, I suggested the addition of some discussion of this violated assumption. The author's respond by indicating that it was actually the 'rule' to violate this assumption and have not added necessary discussion. The idea that violation of this assumption is the 'rule', is simply false. Many studies that focus on recombination acknowledge the fact that phylogenies like those used by the authors assume no recombination, and cautiously interpret results in that context. Furthermore, alternative approaches to networking like a neighbor-net network that allow for recombination are widely used by other studies.

Here, we were simply pointing out that nearly all phylogenetic and molecular dating studies performed using nuclear sequences from eukaryotes violate the assumption of no recombination (with the rare exceptions of parthenogenetic lineages). Yet, for the hundreds of papers we've read with phylogenetic trees in them, we don't ever recall reading any language that states something to the effect: "extreme care should be made in drawing ANY inferences from the branch lengths and branch orders in this tree because formal tests for admixture have not been performed". Likewise, we have never seen any phylogenomic dating studies which provide the caveat: "our date estimates could be off by X % or more if any admixture events have occurred in the Y years since our predicted TMRCA."

In my original review I pointed out that the authors inferences of parasexuality are not robust. The crux of their approach is extrapolating from other studies' inferences of the length of meiotic vs mitotic conversions observed in model organisms in vitro. Based on the identification of shorter-length conversions, they infer mitotic mechanisms are more likely and thus invoke parasexuality. This approach is 1) unprecedented and unvetted 2) There is no evidence of this conversion pattern in natural systems where parasexuality is known to occur (e.g., in the best studied natural examples of parasexuality from *Cryphonectria parasitica* there are no reports of this pattern) 3) based on extrapolation of results from entirely different organisms in a controlled laboratory environment. Despite these glaring limitations, the authors present inferences of parasexuality as fact and do not offer any suggestion of doubt surrounding these inferences.

All mentions about the possibility of parasexuality have been eliminated. However, it should be noted that there are no reports of this pattern in studied parasexual systems because nobody has generated the data that would be required to detect it.

Instead of thoughtful discussion of possible alternative hypotheses that readers may consider to explain the results of the study, the authors spend nearly entire discussion lauding their results and wildly extrapolating to other systems. While I pointed out several possible instances where careful discussion of alternative hypotheses would be appropriate, the authors respond by invoking 'Occom's razor' rather than perform additional analyses or adding any discussion. Occom's razor is a useful heuristic to choose a hypotheses, it is not a robust way to falsify a hypothesis or to dismiss alternative hypotheses.

Hopefully, now that we have explained the experiments better in the main text, along with the above list explaining how twelve different experiments/observations all converge on the same conclusion that WB/GLS arose in a multi-hybrid swarm, the reviewer can now see that there is no other reasonable explanation for the data. They will also hopefully recognize that Occum's razor can be applied in this case because, to propose any alternative hypothesis would have to include multiple caveats, and would require ignoring several key observations and/or invoke mechanisms that go against key evolutionary principles. To be more specific, there is no reasonable, alternative explanation for members of the PoT/PoL1 population having genomes that are variable mosaics of chromosome segments, each of which shows greater sequence similarity to a member of a candidate donor population, than exists among different members of that donor population.

For example, the authors do not ever consider the potential that selection may be an important factor in their results, but that selection manifests in recombination load. As a result, the vast majority of recombination events would be selected against, and the inferred events may be rare examples of combinations that are advantageous. This alternative hypothesis marks a fundamentally different explanation to what the authors posit as fact.

We think the reviewer is making the same conclusion that we outlined in our discussion but perhaps we didn't explain our logic carefully enough. It is generally assumed that when pathogens jump hosts, they are initially mal-adapted until new, beneficial variants accumulate and are operated upon by selection. The point we were trying to make (and have hopefully clarified in this revision) is that for WB/GLS, the newly-evolved pathogens appear to have been fully adapted "out-of-the-box" because the diseases exploded onto the scene before there was much time for any new mutations to have occurred. Furthermore, for the very few new SNPs that we do find in present-day samples, a majority are unique to an individual isolate, and therefore show no evidence of having been enriched by selection. In fact,

the only evidence of selection we see is in favor of a null allele of *PWT3* that was inherited as standing variation - this mutation was not needed for the initial host jump but was needed to expand the host range to include *RWT3* wheat. These points are (hopefully) now clarified in the revised text.

Finally, while there does seem to have been some clarification of the language and flow in this document, problems with writing persist. Speculation presented as fact, unscientific language, and grammatical errors are widespread in the MS. Several of these issues were noted in my original review. I am hesitant to spend the time pointing out specifics of these as I highlighted several in my original review that the authors have not addressed. For example, I previously noted that the statement "Our findings point to an overlooked role for happenstance in creating situations that allow organisms to skirt rules that would normally hold evolution in check" is unclear, does not clearly reflect what was found, and incorrectly misrepresents evolutionary theory (I previously pointed out that even Darwin emphasized the importance of genetic drift). This statement has not been fixed, contextualized, or clarified and now appears to occur twice on lines 42, 586.

These points have now been addressed by focusing discussion on the relevance of our findings to host jumps.

Lastly, it should be noted that we changed the colors in the figures to use a schema that should be more "readable" to those with color blindness.

Decision Letter, second revision:

16th May 2023

Dear Mark,

Thank you for submitting your revised manuscript "RECENT CO-EVOLUTION OF TWO PANDEMIC PLANT DISEASES IN A MULTI-HYBRID SWARM" (NATECOLEVOL-211014889B). We have sent your manuscript to a new reviewer and have asked them to comment on the points of disagreement between the authors and Reviewer 2. Based on their comments, copied below, we are happy in principle to publish it in Nature Ecology & Evolution, pending minor revisions to satisfy the reviewers'

36final requests and to comply with our editorial and formatting guidelines. I should stress that we agree with the reviewer that the paper would greatly benefit from a discussion of comparisons to other plant pathogen invasions. We cannot allow for expansion of the Discussion given that the manuscript is already about 5000 words but we strongly encourage you to replace some of the points with a broad discussion as suggested by the reviewer.

[REDACTED]

Reviewer #3 (Remarks to the Author):

Comments

The authors describe a process, that in its simplest form is just recombination. They suggest that recombination among diverse pathogens could account for the rapid emergence of new diseases. It is noteworthy that the authors do this in a pathogen of wheat, but these processes are much broader than the several hosts in this system because they tackle a somewhat dogmatic rationale applied in the plant pathogen community of ignoring recombination and opting to use clonal lineages develop and understanding of host resistance, and pathogen emergence. It is just recombination, but it is key for host jumps and pathogen emergence.

In reading the MS I was also asked to consider previous reviewer's comments. However, I read it first without prejudice from comments. I feel this work is an important step forward and I can see why the community might be resistant. I didn't feel that the authors overstated their assertions.

Below I outline some thoughts on the reviewer's comments and the authors' response.

Reviewer's comments:

The first reviewer appears unhappy that authors "have dismissed all of my suggestions to further support their claims". I can't see all of those suggestions as they were in a former revision and so can't comment as to what extent they have been addressed. Only that the authors present their hypothesis/model for introgression and emergence. This model must be published in order that other researchers are able to test it, or debate it. From what I can see though:

- I'm not sure what adding Rice blast would add to this MS. I think it would distract from the main message but would probably be worth doing in follow up work. Even if it is just to say that it wasn't part of the swarm. Perhaps work on a broader phylogeny might highlight the preponderance for this process to operate in this part of the tree of life.
- I'm not sure what analysis using ClonalFrameML would add to this work. The authors have gone to great lengths to construct their BEAST tree from more stable regions of the genome, and they have

37also painted all the recombination. It's quite clear, and nice, that they highlight early isolates have fewer contributors. It is often too easy, as a reviewer, to suggest another piece of software but really I think it's the reviewer's job to suggest what it is about the current methodology that would make the authors conclusions erroneous.

- ARGweaver is apparently inappropriate and yet the authors did try it.
- I'm not going to continue about software.

Phylogenies do violate expectations because of recombination, but the authors do also use a SplitsTree network. The authors have identified regions of the genome that are more appropriate for bifurcation and it's true that you don't read a disclaimer on every other tree that ignores these issues. In fact, the whole ethos of this paper is that one should consider recombination.

To the point about how authors spend "entire discussion lauding their results", I can see the reviewers point here somewhat. I'm not sure how close the authors are to their word count but I think they would increase the broader impact of their work if they were to be bolder in their comparisons to other (plant) pathogen invasions. This would allow those readers from outside of this skillset to be able to extrapolate the understanding to their own system. I don't work on this system and did find the detailed explanation of the different events and the explanation of genes that needed to be switched off in order to infect new hosts, quite detailed (Discussion). I'm sure it's critical for this system but I think there is also benefit in broadening things. From memory some places to start might include but not limited to:

- Stukenbrock, E. H., Christiansen, F. B., Hansen, T. T., Dutheil, J. Y. & Schierup, M. H. (2012). Fusion of Two Divergent Fungal Individuals Led to the Recent Emergence of a Unique Widespread Pathogen Species. *Proc.Natl. Acad. Sci. USA*, 109(27), 10954. doi:10.1073/pnas.1201403109
 - o Work which they already cite in the introduction and would close the loop
- Brasier CM, Kirk SA. Rapid emergence of hybrids between the two subspecies of *Ophiostoma novo-ulmi* with a high level of pathogenic fitness. *Plant Pathol.* 2010;59(1):186–99.
 - o Work which might be used to highlight the devastation caused by recombination/hybridisation.
- McMullan M, Gardiner A, Bailey K, Kemen E, Ward BJ, Cevik V, et al. Evidence for suppression of immunity as a driver for genomic introgressions and host range expansion in races of *Albugo candida*, a generalist parasite. *Elife* [Internet]. 2015 Feb 27;4:1–24. Available from: <http://elifesciences.org/lookup/doi/10.7554/eLife.04550>
 - o Similar to the present MS but more limited in data. Broadens the scope by including an oomycete as well as dealing with the difficulties of how divergent pathogens cross
- Galaway F, Yu R, Constantinou A, Prugnolle F, Wright GJ. Resurrection of the ancestral RH5 invasion ligand provides a molecular explanation for the origin of *P. falciparum* malaria in humans. *PLoS Biol* [Internet]. 2019;17(10):e3000490. Available from: <http://www.ncbi.nlm.nih.gov/pubmed/31613878>
 - o Perhaps a stretch but brings in zoonoses and humans with a focus on the impact of a specific gene.
- All just suggestions to give a flavour of what the discussion could do

My comments

There are no fundamental flaws in this work. There is a description of observation and a model which can be tested by the community.

Minor comments

Ln48 "Most new plant diseases emerge when existing pathogen populations overcome prior barriers to infection and colonize former non-hosts". Is this tautology? What other kinds of disease emergence are there? Maybe a citation so that I can go and find out about this?

Ln140 "B71 often showed perfect haplotype identity", in what proportion of windows?

Ln142 "Winning" might be better as "source", but it's the author's decision.

Figure 1, where is B71 on your network?

Ln259 This is the first time you mention "admixture 1" and then later admixture 2 below. This is confusing because the point of reference here is figure 5 and there is no mention of this until figure 6. It might be worth stating early in this section that the date estimates generated from the BEAST tree (fig5) are used in a graphical model (fig6) that describes the invasion swarm/hybridisation hypothesis. --I think perhaps that stating that this is a hypothesis based on all the available evidence might reduce the negativity of other reviewers. I think these figures (5&6) go well together and the authors could tie them together further by adding a reference to key points (from fig6 -perhaps Admix1,2 etc) as symbols to the timeline on the beast tree (fig5). Making it clear that these are predictions based on the divergence times generated by that tree and perhaps referring to fig 6 in the legend.

Ln264 I'm not sure what in 5b tells me in relation to this sentence. How does 5b relate to those dates?

Comments the authors may wish to ignore:

I think the authors perhaps miss a trick in their discussion on the hopeful monster and expectations of host jump adaptation taking a long time. This expectation surely assumes that the host is both diverse and evolving. This is not the case in crops.

It might have been nice to see if there was a relationship between the regions of the genome (not) impacted by recombination and the genes found there. It should at least appear in follow up work that includes gene function and diversity, within and among clades (presumably).

Our ref: NATECOLEVOL-211014889B

6th June 2023

Dear Dr. Farman,

Thank you for your patience as we've prepared the guidelines for final submission of your Nature Ecology & Evolution manuscript, "RECENT CO-EVOLUTION OF TWO PANDEMIC PLANT DISEASES IN A MULTI-HYBRID SWARM" (NATECOLEVOL-211014889B). Please carefully follow the step-by-step instructions provided in the attached file, and add a response in each row of the table to indicate the changes that you have made. Please also check and comment on any additional marked-up edits we have proposed within the text. Ensuring that each point is addressed will help to ensure that your revised manuscript can be swiftly handed over to our production team.

****We would like to start working on your revised paper, with all of the requested files and forms, as soon as possible (preferably within two weeks). Please get in contact with us immediately if you anticipate it taking more than two weeks to submit these revised files.****

In recognition of the time and expertise our reviewers provide to Nature Ecology & Evolution's editorial process, we would like to formally acknowledge their contribution to the external peer review of your manuscript entitled "RECENT CO-EVOLUTION OF TWO PANDEMIC PLANT DISEASES IN A MULTI-HYBRID SWARM". For those reviewers who give their assent, we will be publishing their names alongside the published article.

Nature Ecology & Evolution offers a Transparent Peer Review option for new original research manuscripts submitted after December 1st, 2019. As part of this initiative, we encourage our authors to support increased transparency into the peer review process by agreeing to have the reviewer comments, author rebuttal letters, and editorial decision letters published as a Supplementary item. When you submit your final files please clearly state in your cover letter whether or not you would like to participate in this initiative. Please note that failure to state your preference will result in delays in accepting your manuscript for publication.

Cover suggestions

As you prepare your final files we encourage you to consider whether you have any images or illustrations that may be appropriate for use on the cover of Nature Ecology & Evolution.

Covers should be both aesthetically appealing and scientifically relevant, and should be supplied at the

40best quality available. Due to the prominence of these images, we do not generally select images featuring faces, children, text, graphs, schematic drawings, or collages on our covers.

Nature Ecology & Evolution has now transitioned to a unified Rights Collection system which will allow our Author Services team to quickly and easily collect the rights and permissions required to publish your work. Approximately 10 days after your paper is formally accepted, you will receive an email in providing you with a link to complete the grant of rights. If your paper is eligible for Open Access, our Author Services team will also be in touch regarding any additional information that may be required to arrange payment for your article.

Please note that *Nature Ecology & Evolution* is a Transformative Journal (TJ). Authors may publish their research with us through the traditional subscription access route or make their paper immediately open access through payment of an article-processing charge (APC). Authors will not be required to make a final decision about access to their article until it has been accepted. [Find out more about Transformative Journals](https://www.springernature.com/gp/open-research/transformative-journals)

Authors may need to take specific actions to achieve [compliance](https://www.springernature.com/gp/open-research/funding/policy-compliance-faqs) with funder and institutional open access mandates. If your research is supported by a funder that requires immediate open access (e.g. according to [Plan S principles](https://www.springernature.com/gp/open-research/plan-s-compliance)) then you should select the gold OA route, and we will direct you to the compliant route where possible. For authors selecting the subscription publication route, the journal's standard licensing terms will need to be accepted, including [self-archiving-and-license-to-publish](https://www.nature.com/nature-portfolio/editorial-policies/self-archiving-and-license-to-publish). Those licensing terms will supersede any other terms that the author or any third party may assert apply to any version of the manuscript.

[REDACTED]

[REDACTED]

Reviewer #3:

Remarks to the Author:

Comments

The authors describe a process, that in its simplest form is just recombination. They suggest that recombination among diverse pathogens could account for the rapid emergence of new diseases. It is noteworthy that the authors do this in a pathogen of wheat, but these processes are much broader than the several hosts in this system because they tackle a somewhat dogmatic rationale applied in the plant pathogen community of ignoring recombination and opting to use clonal lineages develop and understanding of host resistance, and pathogen emergence. It is just recombination, but it is key for host jumps and pathogen emergence.

In reading the MS I was also asked to consider previous reviewer's comments. However, I read it first without prejudice from comments. I feel this work is an important step forward and I can see why the community might be resistant. I didn't feel that the authors overstated their assertions.

Below I outline some thoughts on the reviewer's comments and the authors' response.

Reviewer's comments:

The first reviewer appears unhappy that authors "have dismissed all of my suggestions to further support their claims". I can't see all of those suggestions as they were in a former revision and so can't comment as to what extent they have been addressed. Only that the authors present their hypothesis/model for introgression and emergence. This model must be published in order that other researchers are able to test it, or debate it. From what I can see though:

- I'm not sure what adding Rice blast would add to this MS. I think it would distract from the main message but would probably be worth doing in follow up work. Even if it is just to say that it wasn't part of the swarm. Perhaps work on a broader phylogeny might highlight the preponderance for this process to operate in this part of the tree of life.
- I'm not sure what analysis using ClonalFrameML would add to this work. The authors have gone to great lengths to construct their BEAST tree from more stable regions of the genome, and they have also painted all the recombination. It's quite clear, and nice, that they highlight early isolates have fewer contributors. It is often too easy, as a reviewer, to suggest another piece of software but really I think it's the reviewer's job to suggest what it is about the current methodology that would make the authors conclusions erroneous.

42- ARGweaver is apparently inappropriate and yet the authors did try it.
- I'm not going to continue about software.

Phylogenies do violate expectations because of recombination, but the authors do also use a SplitsTree network. The authors have identified regions of the genome that are more appropriate for bifurcation and it's true that you don't read a disclaimer on every other tree that ignores these issues. In fact, the whole ethos of this paper is that one should consider recombination.

To the point about how authors spend "entire discussion lauding their results", I can see the reviewers point here somewhat. I'm not sure how close the authors are to their word count but I think they would increase the broader impact of their work if they were to be bolder in their comparisons to other (plant) pathogen invasions. This would allow those readers from outside of this skillset to be able to extrapolate the understanding to their own system. I don't work on this system and did find the detailed explanation of the different events and the explanation of genes that needed to be switched off in order to infect new hosts, quite detailed (Discussion). I'm sure it's critical for this system but I think there is also benefit in broadening things. From memory some places to start might include but not limited to:

- Stukenbrock, E. H., Christiansen, F. B., Hansen, T. T., Duthiel, J. Y. & Schierup, M. H. (2012). Fusion of Two Divergent Fungal Individuals Led to the Recent Emergence of a Unique Widespread Pathogen Species. *Proc. Natl. Acad. Sci. USA*, 109(27), 10954. doi:10.1073/pnas.1201403109
 - o Work which they already cite in the introduction and would close the loop
- Brasier CM, Kirk SA. Rapid emergence of hybrids between the two subspecies of *Ophiostoma novoulmi* with a high level of pathogenic fitness. *Plant Pathol.* 2010;59(1):186–99.
 - o Work which might be used to highlight the devastation caused by recombination/hybridisation.
- McMullan M, Gardiner A, Bailey K, Kemen E, Ward BJ, Cevik V, et al. Evidence for suppression of immunity as a driver for genomic introgressions and host range expansion in races of *Albugo candida*, a generalist parasite. *Elife* [Internet]. 2015 Feb 27;4:1–24. Available from: <http://elifesciences.org/lookup/doi/10.7554/eLife.04550>
 - o Similar to the present MS but more limited in data. Broadens the scope by including an oomycete as well as dealing with the difficulties of how divergent pathogens cross
- Galaway F, Yu R, Constantinou A, Prugnolle F, Wright GJ. Resurrection of the ancestral RH5 invasion ligand provides a molecular explanation for the origin of *P. falciparum* malaria in humans. *PLoS Biol* [Internet]. 2019;17(10):e3000490. Available from: <http://www.ncbi.nlm.nih.gov/pubmed/31613878>
 - o Perhaps a stretch but brings in zoonoses and humans with a focus on the impact of a specific gene.
- All just suggestions to give a flavour of what the discussion could do

My comments

There are no fundamental flaws in this work. There is a description of observation and a model which can be tested by the community.

Minor comments

In48 "Most new plant diseases emerge when existing pathogen populations overcome prior barriers to infection and colonize former non-hosts". Is this tautology? What other kinds of disease emergence are there? Maybe a citation so that I can go and find out about this?

43Ln140 "B71 often showed perfect haplotype identity", in what proportion of windows?
Ln142 "Winning" might be better as "source", but it's the author's decision.

Figure 1, where is B71 on your network?

Ln259 This is the first time you mention "admixture 1" and then later admixture 2 below. This is confusing because the point of reference here is figure 5 and there is no mention of this until figure 6. It might be worth stating early in this section that the date estimates generated from the BEAST tree (fig5) are used in a graphical model (fig6) that describes the invasion swarm/hybridisation hypothesis. --I think perhaps that stating that this is a hypothesis based on all the available evidence might reduce the negativity of other reviewers. I think these figures (5&6) go well together and the authors could tie them together further by adding a reference to key points (from fig6 -perhaps Admix1,2 etc) as symbols to the timeline on the beast tree (fig5). Making it clear that these are predictions based on the divergence times generated by that tree and perhaps referring to fig 6 in the legend.

Ln264 I'm not sure what in 5b tells me in relation to this sentence. How does 5b relate to those dates?

Comments the authors may wish to ignore:

I think the authors perhaps miss a trick in their discussion on the hopeful monster and expectations of host jump adaptation taking a long time. This expectation surely assumes that the host is both diverse and evolving. This is not the case in crops.

It might have been nice to see if there was a relationship between the regions of the genome (not impacted by recombination and the genes found there. It should at least appear in follow up work that includes gene function and diversity, within and among clades (presumably).

Author Rebuttal, second revision:

Reviewer #3:

Remarks to the Author:

Comments

The authors describe a process, that in its simplest form is just recombination. They suggest that recombination among diverse pathogens could account for the rapid emergence of new diseases. It is noteworthy that the authors do this in a pathogen of wheat, but these processes are much broader than the several hosts in this system because they tackle a somewhat dogmatic rationale applied in the plant pathogen community of ignoring recombination and opting to use clonal lineages develop and understanding of host resistance, and pathogen emergence. It is just recombination, but it is key for host jumps and pathogen emergence.

In reading the MS I was also asked to consider previous reviewer's comments. However, I read it first without prejudice from comments. I feel this work is an important step forward and I can see why the community might be

44resistant. I didn't feel that the authors overstated their assertions.

Below I outline some thoughts on the reviewer's comments and the authors' response.

Reviewer's comments:

The first reviewer appears unhappy that authors "have dismissed all of my suggestions to further support their claims". I can't see all of those suggestions as they were in a former revision and so can't comment as to what extent they have been addressed. Only that the authors present their hypothesis/model for introgression and emergence. This model must be published in order that other researchers are able to test it, or debate it. From what I can see though:

- I'm not sure what adding Rice blast would add to this MS. I think it would distract from the main message but would probably be worth doing in follow up work. Even if it is just to say that it wasn't part of the swarm. Perhaps work on a broader phylogeny might highlight the preponderance for this process to operate in this part of the tree of life.
- I'm not sure what analysis using ClonalFrameML would add to this work. The authors have gone to great lengths to construct their BEAST tree from more stable regions of the genome, and they have also painted all the recombination. It's quite clear, and nice, that they highlight early isolates have fewer contributors. It is often too easy, as a reviewer, to suggest another piece of software but really I think it's the reviewer's job to suggest what it is about the current methodology that would make the authors conclusions erroneous.
- ARGweaver is apparently inappropriate and yet the authors did try it.
- I'm not going to continue about software.

Phylogenies do violate expectations because of recombination, but the authors do also use a SplitsTree network. The authors have identified regions of the genome that are more appropriate for bifurcation and it's true that you don't read a disclaimer on every other tree that ignores these issues. In fact, the whole ethos of this paper is that one should consider recombination.

We are very pleased that the reviewer recognizes the importance of using non-recombined for coalescence analyses.

To the point about how authors spend "entire discussion lauding their results", I can see the reviewers point here somewhat. I'm not sure how close the authors are to their word count but I think they would increase the broader impact of their work if they were to be bolder in their comparisons to other (plant) pathogen invasions. This would allow those readers from outside of this skillset to be able to extrapolate the understanding to their own system. I don't work on this system and did find the detailed explanation of the different events and the explanation of genes that needed to be switched off in order to infect new hosts, quite detailed (Discussion). I'm sure it's critical for this system but I think there is also benefit in broadening things. From memory some places to start might include but not limited to:

First, we'd like to point out that the word limit meant that we were unable to broaden the discussion as much as we'd have liked because earlier reviews made it clear that we needed to do a much better job of walking the readers through the logic underpinning the multi-swarm model. Then, our intention was to focus attention mainly on the insights we gained into new disease emergence via host jumps. With this in mind, we felt it important to discuss the following points:

- 1) Wheat blast/gray leaf spot evolution appears to have been precipitated by the introduction of *Urochloa* into Brazil in 1950s.
- 2) This is the first example of a new "hybrid" disease where genes that determine specificity on a new host have been cloned and, therefore, allow us to provide a mechanistic explanation of how hybridization drove the host jump, thereby filling a gap in hybridization  host range expansion literature.

45- 3) Equally importantly, our data imply that the genes were recombined in a pre-adaptive fashion and not according to the prevailing spring-board + fine-tuning model (Inoue et al. 2017; Asume et al. 2021). We feel this is important to point out because the model is intuitively appealing (and has been the subject of a number of reviews) but is not really supported by the phylogenomic data.
- 4) One thing that sets our work apart from previous studies and, we believe, deserves discussion is that wheat blast and gray leaf spot evolved so recently that we can estimate the number of new SNPs that have arisen in each isolate since the founder event. These numbers are so few that they pale into insignificance when compared with the divergence generated by recombination and argue against a role for post-jump adaptation in disease establishment.
- 5) Finally, we wanted to emphasize the emerging pattern where all of the recently emerged pathogen populations for which phylogenomic data are available (four at present: WB, GLS, *Blumeria graminis tritici* and *Zymoseptoria pseudotritici*) have hybrid swarm origins, exhibit unexpectedly high genetic diversity, and were so quickly established in their new hosts that there wasn't much time for new mutations to occur. This does not fit the widely accepted "host jump speciation" model and suggests that recombination alone can instantaneously generate aggressive pathogens with new host specificities.

Having said this, we value the reviewer's suggestions and have incorporated some of the suggested talking points into the revised discussion:

• Stukenbrock, E. H., Christiansen, F. B., Hansen, T. T., Dutheil, J. Y. & Schierup, M. H. (2012). Fusion of Two Divergent Fungal Individuals Led to the Recent Emergence of a Unique Widespread Pathogen Species. *Proc. Natl. Acad. Sci. USA*, 109(27), 10954. doi:10.1073/pnas.1201403109

This work was discussed in the previous submission and we continue to use it as a prior example of a hybrid swarm generating a highly diverse pathogen population that emerged so quickly that it implies preadaptation; and that recombinational load/epistasis doesn't appear to hinder establishment on the new hosts.

o Work which they already cite in the introduction and would close the loop

• Brasier CM, Kirk SA. Rapid emergence of hybrids between the two subspecies of *Ophiostoma novo-ulmi* with a high level of pathogenic fitness. *Plant Pathol.* 2010;59(1):186–99.

o Work which might be used to highlight the devastation caused by recombination/hybridisation.

We reference this and other papers as examples of how hybridization can lead to disease re-emergence.

• McMullan M, Gardiner A, Bailey K, Kemen E, Ward BJ, Cevik V, et al. Evidence for suppression of immunity as a driver for genomic introgressions and host range expansion in races of *Albugo candida*, a generalist parasite. *Elife* [Internet]. 2015 Feb 27;4:1–24. Available from: <http://elifesciences.org/lookup/doi/10.7554/eLife.04550>

o Similar to the present MS but more limited in data. Broadens the scope by including an oomycete as well as dealing with the difficulties of how divergent pathogens cross

We reference this system as an example of recombination driving host range expansion and discuss and to illustrate a potential mechanism by which divergently adapted strains could have come together in a way that allowed mating to occur.

• Galaway F, Yu R, Constantinou A, Prugnolle F, Wright GJ. Resurrection of the ancestral RH5 invasion ligand provides a molecular explanation for the origin of *P. falciparum* malaria in humans. *PLoS Biol* [Internet]. 2019;17(10):e3000490. Available from: <http://www.ncbi.nlm.nih.gov/pubmed/31613878>

- o Perhaps a stretch but brings in zoonoses and humans with a focus on the impact of a specific gene.
- All just suggestions to give a flavour of what the discussion could do

Due to space limitations, we are unable to broaden the scope of the discussion this far.

My comments

There are no fundamental flaws in this work. There is a description of observation and a model which can be tested by the community.

Minor comments

Ln48 "Most new plant diseases emerge when existing pathogen populations overcome prior barriers to infection and colonize former non-hosts". Is this tautology? What other kinds of disease emergence are there? Maybe a citation so that I can go and find out about this?

We see the reviewer's point here. However, new diseases can also occur when non-pathogenic organisms gain infection capability. Having said this, our intention was to introduce the idea that pathogen populations specialized on one host must overcome former genetic barriers to infection before they can become established in another host. This point has now been clarified.

Ln140 "B71 often showed perfect haplotype identity", in what proportion of windows?

Fifty-two to 92 % of windows (data now included in text and relevant script included in GitHub)

Ln142 "Winning" might be better as "source", but it's the author's decision.

Changed to "most related" isolate

Figure 1, where is B71 on your network?

B71 was not included in these analyses but is essentially clonal to all PoT32 haplotype isolates (average ~300 SNPs genome-wide). Its presence in the trees/network is not really important in relation to the conclusions drawn. However, its inferred position (based on genome-wide divergence) is now noted in the legend for the benefit of those with specific interest in this strain.

Ln259 This is the first time you mention "admixture 1" and then later admixture 2 below. This is confusing because the point of reference here is figure 5 and there is no mention of this until figure 6. It might be worth stating early in this section that the date estimates generated from the BEAST tree (fig5) are used in a graphical model (fig6) that describes the invasion swarm/hybridisation hypothesis.--I think perhaps that stating that this is a hypothesis based on all the available evidence might reduce the negativity of other reviewers.

We have incorporated these excellent suggestions

I think these figures (5&6) go well together and the authors could tie them together further by adding a reference to key points (from fig6 -perhaps Admix1,2 etc) as symbols to the timeline on the beast tree (fig5). Making it clear that these are predictions based on the divergence times generated by that tree and perhaps referring to fig 6 in the legend.

This was another very helpful suggestion and appropriate changes have been incorporated into the revision.

Ln264 I'm not sure what in 5b tells me in relation to this sentence. How does 5b relate to those dates?

47This was a carryover from when the figure had only two panels. Error corrected

Comments the authors may wish to ignore:

I think the authors perhaps miss a trick in their discussion on the hopeful monster and expectations of host jump adaptation taking a long time. This expectation surely assumes that the host is both diverse and evolving. This is not the case in crops.

We now raise this point in the penultimate sentence of the discussion.

It might have been nice to see if there was a relationship between the regions of the genome (not) impacted by recombination and the genes found there. It should at least appear in follow up work that includes gene function and diversity, within and among clades (presumably).

We certainly plan to investigate the gene content of invariant regions of the genome. However, this is beyond the scope of the present work and word limits prevented us even from discussing the known AVR gene alleles in any depth. It is also important to recognize that the lack of allelic diversity across large portions of the wheat/PoL genomes may also be due to a meiotic drive phenomenon because it is hard to imagine how whole chromosome regions would otherwise be counter-selected.

Final Decision Letter:

28th September 2023

Dear Mark,

We are pleased to inform you that your Article entitled "RECENT CO-EVOLUTION OF TWO PANDEMIC PLANT DISEASES IN A MULTI-HYBRID SWARM", has now been accepted for publication in Nature Ecology & Evolution.

Over the next few weeks, your paper will be copyedited to ensure that it conforms to Nature Ecology and Evolution style. Once your paper is typeset, you will receive an email with a link to choose the appropriate publishing options for your paper and our Author Services team will be in touch regarding any additional information that may be required

Due to the importance of these deadlines, we ask you please us know now whether you will be difficult to contact over the next month. If this is the case, we ask you provide us with the contact information (email, phone and fax) of someone who will be able to check the proofs on your behalf, and who will be available to address any last-minute problems . Once your paper has been scheduled for online publication, the Nature press office will be in touch to confirm the details.

48Acceptance of your manuscript is conditional on all authors' agreement with our publication policies (see www.nature.com/authors/policies/index.html). In particular your manuscript must not be published elsewhere and there must be no announcement of the work to any media outlet until the publication date (the day on which it is uploaded onto our web site).

Please note that *Nature Ecology & Evolution* is a Transformative Journal (TJ). Authors may publish their research with us through the traditional subscription access route or make their paper immediately open access through payment of an article-processing charge (APC). Authors will not be required to make a final decision about access to their article until it has been accepted. [Find out more about Transformative Journals](https://www.springernature.com/gp/open-research/transformative-journals)

Authors may need to take specific actions to achieve [compliance with funder and institutional open access mandates](https://www.springernature.com/gp/open-research/funding/policy-compliance-faqs). If your research is supported by a funder that requires immediate open access (e.g. according to [Plan S principles](https://www.springernature.com/gp/open-research/plan-s-compliance)) then you should select the gold OA route, and we will direct you to the compliant route where possible. For authors selecting the subscription publication route, the journal's standard licensing terms will need to be accepted, including [those licensing terms](https://www.nature.com/nature-portfolio/editorial-policies/self-archiving-and-license-to-publish) will supersede any other terms that the author or any third party may assert apply to any version of the manuscript.

We welcome the submission of potential cover material (including a short caption of around 40 words) related to your manuscript; suggestions should be sent to Nature Ecology & Evolution as electronic files (the image should be 300 dpi at 210 x 297 mm in either TIFF or JPEG format). Please note that such pictures should be selected more for their aesthetic appeal than for their scientific content, and that colour images work better than black and white or grayscale images. Please do not try to design a cover with the Nature Ecology & Evolution logo etc., and please do not submit composites of images related to your work. I am sure you will understand that we cannot make any promise as to whether any of your suggestions might be selected for the cover of the journal.

49You can now use a single sign-on for all your accounts, view the status of all your manuscript submissions and reviews, access usage statistics for your published articles and download a record of your refereeing activity for the Nature journals.

You can generate the link yourself when you receive your article DOI by entering it here: <http://authors.springernature.com/share>.

[REDACTED]

P.S. Click on the following link if you would like to recommend Nature Ecology & Evolution to your librarian <http://www.nature.com/subscriptions/recommend.html#forms>

** Visit the Springer Nature Editorial and Publishing website at http://editorial-jobs.springernature.com?utm_source=ejp_NEcoE_email&utm_medium=ejp_NEcoE_email&utm_campaign=ejp_NEcoE for more information about our career opportunities. If you have any questions please click [here](mailto:editorial.publishing.jobs@springernature.com). **